# MYCN-driven fatty acid uptake is a metabolic vulnerability in neuroblastoma

Ling Tao [1,2], Mahmoud A. Mohammad [3,4], Giorgio Milazzo[5], Myrthala Moreno-Smith[1,2], Tajhal D. Patel[1], Barry Zorman[1,2], Andrew Badachhape[6], Blanca E. Hernandez [1,2], Amber B. Wolf [1,2], Zihua Zeng[7], Jennifer H. Foster[1], Sara Aloisi[5], Pavel Sumazin[1,2], Youli Zu[7], John Hicks[8], Ketan B. Ghaghada[6], Nagireddy Putluri[2,9,10], Giovanni Perini[5], Cristian Coarfa [2,10] & Eveline Barbieri [1,2✉]

Neuroblastoma (NB) is a childhood cancer arising from sympatho-adrenal neural crest cells. *MYCN* amplification is found in half of high-risk NB patients; however, no available therapies directly target MYCN. Using multi-dimensional metabolic profiling in MYCN expression systems and primary patient tumors, we comprehensively characterized the metabolic landscape driven by MYCN in NB. *MYCN* amplification leads to glycerolipid accumulation by promoting fatty acid (FA) uptake and biosynthesis. We found that cells expressing amplified *MYCN* depend highly on FA uptake for survival. Mechanistically, MYCN directly upregulates FA transport protein 2 (FATP2), encoded by *SLC27A2*. Genetic depletion of *SLC27A2* impairs NB survival, and pharmacological *SLC27A2* inhibition selectively suppresses tumor growth, prolongs animal survival, and exerts synergistic anti-tumor effects when combined with conventional chemotherapies in multiple preclinical NB models. This study identifies FA uptake as a critical metabolic dependency for *MYCN*-amplified tumors. Inhibiting FA uptake is an effective approach for improving current treatment regimens.

[1] Department of Pediatrics, Section of Hematology-Oncology, Texas Children's Cancer and Hematology Centers, Baylor College of Medicine, Houston, TX 77030, USA. [2] Dan L Duncan Comprehensive Cancer Center, Baylor College of Medicine, Houston, TX 77030, USA. [3] Department of Pediatrics-Nutrition, Baylor College of Medicine, Houston, TX 77030, USA. [4] Food Science and Nutrition Department, National Research Centre, El-Buhouth St., Dokki, Cairo 12622, Egypt. [5] Department of Pharmacy and Biotechnology, University of Bologna, Bologna 40126, Italy. [6] Department of Radiology, Texas Children's Hospital, Baylor College of Medicine, Houston, TX 77030, USA. [7] Department of Pathology and Genomic Medicine, Houston Methodist Hospital, Houston, TX 77030, USA. [8] Department of Pathology and Immunology, Baylor College of Medicine, Houston, TX 77030, USA. [9] Advanced Technology Core, Baylor College of Medicine, Houston, TX 77030, USA. [10] Department of Molecular and Cellular Biology, Baylor College of Medicine, Houston, TX 77030, USA. ✉email: exbarbie@txch.org

Neuroblastoma (NB), a childhood solid tumor of the sympathetic nervous system, accounts for 15% of total childhood cancer mortality[1]. Despite multimodal therapies, the 5-year survival rate for high-risk patients with NB remains below 50%[1]. Almost half of high-risk NB patients harbor the amplified MYCN oncogene, the primary oncogenic driver of NB, leading to resistance to therapy and poor clinical outcomes[2,3]. Pleiotropic effects and the lack of enzymatic activity[4] make directly targeting the MYCN oncoprotein challenging. Therefore, new strategies for disrupting MYCN oncogenic programming are critical for developing effective NB therapies. MYCN facilitates NB growth in complex microenvironments with limited nutrient availability by overriding hypoxia-inducible factor (HIF1α)-mediated cell cycle arrest and promoting intratumoral vascular development to facilitate nutrient access by interior tumor regions[5,6]. However, the mechanisms underlying the MYCN reprogramming of NB tumors that permit nutrient recruitment from the microenvironment remain unknown.

Compared with normal cells, tumor cells are metabolically reprogrammed to promote biomass and energy production to support rapid cell growth[7]. These metabolic changes are driven by oncogenic stimuli (e.g. phosphatidyl inositol 3-kinase [PI3K]–AKT–mechanistic target of rapamycin complex 1 [mTORC1][8,9], MYC[10], and RAS[11]), loss of tumor suppressors (p53)[12], and metabolic enzyme dysregulation (e.g. isocitrate dehydrogenase 1, IDH1)[13]. In tumors with activated MYC, increased growth demand is sustained through metabolic reprogramming[14], including stimulation of glucose and glutamine consumption[15,16], mitochondrial biogenesis[17], and biosynthesis of lipids, proteins, and nucleotides[18–20]. MYCN appears to exert similar metabolic functions, promoting glycolysis[21], lipogenesis[22,23], and metabolism of glutamine[21,24], serine[25], and polyamine[26] to enhance macromolecular biosynthesis and energy production. However, we lack a comprehensive understanding of how MYCN-amplified tumors rewire their metabolism and how to incorporate these findings into novel treatment modalities. To address these knowledge gaps, we comprehensively characterized the metabolic landscape driven by MYCN in primary patient samples and multiple in vitro inducible systems.

Lipids play essential roles in supporting tumor survival as membrane components, energy reservoirs, and signaling mediators[27]. Glycerolipids, such as diacylglycerols (DGs), function as secondary messengers that directly bind protein kinase C, RAS guanyl nucleotide-releasing protein, and chimaerins to modulate downstream oncogenic signaling[28]. DGs and fatty acids (FAs) assemble into triacylglycerols (TGs), which are stored in lipid droplets (LDs) as a FA reservoir. Under nutrient deprivation conditions, TGs are hydrolyzed, releasing FAs to maintain lipid homeostasis and meet energy requirements (via β-oxidation)[29]. Cancer cells actively obtain FAs through endogenous biosynthesis and uptake from the microenvironment[27]. Endogenous lipogenic mechanisms include de novo synthesis from acetyl-CoA, FA desaturation, and elongation. Numerous studies support a role for de novo lipogenesis in oncogenesis[27,30]. We and others have shown that MYC or MYCN [MYC(N)] promotes FA synthesis by cooperating with Mondo A–sterol regulatory element-binding protein 1 (SREBP1) or directly activating transcription of key lipogenic enzymes, including acetyl-CoA carboxylase (ACC), fatty acid synthase (FASN), and stearoyl-CoA desaturase (SCD1)[18,23,31]. Recently developed FA synthesis inhibitors (e.g., ACC inhibitor ND-646[32]; FASN inhibitors TVB-2640 and orlistat[22,33]; and SCD1 inhibitor A939572[34]) show varying degrees of anti-tumor activity in preclinical models. However, few have progressed to clinical trials, including TVB-2640 (NCT03808558, NCT03179904, and NCT03032484) and orlistat (non-cancer trials). The limited clinical efficacy of these inhibitors

may be due to limited compound specificity or the activation of compensatory FA uptake mechanisms mediated by oncogenic signaling, such as HIF1α[35], mTOR[36], and RAS[11]. FAs are imported by the membrane CD36 FA translocase and FA transport proteins (FATP1–6, encoded by SLC27A1–6) and trafficked intracellularly via FA binding proteins (FABP1–12)[37]. Emerging evidence supports a role for FA transporters in oncogenesis and chemoresistance. CD36 promotes prostate cancer growth and oral carcinoma metastases[38,39], and drives acquired resistance to lapatinib in HER2-postive breast cancer[40]. FATP1 (SLC27A1) accelerates melanoma initiation, and FATP2 (SLC27A2) confers melanoma resistance to BRAF and MEK inhibition by promoting FA uptake from the microenvironment[41,42]. However, the mechanisms underlying MYC(N) regulation of FA transport in proliferation, disease progression, and resistance to therapy remain unclear.

Using untargeted metabolomics in both in vitro models and patient samples, we characterized the metabolic landscape driven by MYCN in NB and identified key metabolic nodes relevant to NB patients and suitable for therapeutic intervention. We found that MYCN promotes glycerolipid accumulation, a finding validated across multiple systems and supported by targeted lipidomic studies. FAs are required for glycerolipid synthesis. We found that MYCN enhances FA uptake to support cell survival. Mechanistically, we identified the FATP2-encoding SLC27A2 as a novel MYCN transcriptional target required for NB growth. Inhibiting FA uptake by targeting SLC27A2 blocked tumor growth, prolonged animal survival, and enhanced the efficacy of conventional chemotherapy in multiple preclinical NB models. This metabolic dependency identified in MYCN-driven NBs suggests that targeting FA uptake represents a promising strategy for improving current therapies in high-risk disease.

## Results

**MYCN reprograms NB metabolism and promotes glycerolipid accumulation.** MYCN promotes various metabolic adaptations to support tumor growth[21–26,43,44]. However, whether these adaptations are directly relevant to NB patients or can be targeted for therapeutic interventions remains incompletely understood. To address this knowledge gap, we performed untargeted metabolomics (Metabolon Inc., Discovery HD4™ platform) in two NB cell lines with perturbed MYCN expression (LAN5 shMYCN and MYCN3 Tet-On) and NB primary tumors (MYCN-amplified [MNA] n = 18; non MYCN-amplified [non MNA] n = 18, Supplementary Data 1). In the shMYCN model, MYCN was conditionally knocked down (KD) using small hairpin RNA (shRNA)-mediated gene silencing (1 μg/mL doxycycline [DOX] 0–96 h) in MNA LAN5 cells. In the MYCN3 Tet-On model (MYCN-ON), MYCN was conditionally turned on (1 μg/mL DOX, 0–72 h) in non-MNA SHEP sub-cloned cells (Fig. 1a, top, and Supplementary Fig. 1a, left). In total, 580 and 789 metabolites were analyzed in cells and primary tumors, respectively (Supplementary Data 2). MYCN-induced metabolic alterations across different models are presented in pathway classification networks (Fig. 1b). Blue and red circles indicate significantly downregulated and upregulated metabolites ($p \leq 0.05$), respectively. Circle sizes represent the absolute $\log_2$ fold change (FC). Notably, MYCN differentially altered metabolite levels in lipid, amino acid, carbohydrate, and nucleotide super pathways ($p \leq 0.05$), with lipid metabolism as the most represented category (>35% differential metabolites, median $\log_2FC > 0.56$, Supplementary Fig. 1a, right and bottom). These findings suggest that MYCN globally reprograms NB metabolism, particularly lipid metabolism.

To identify metabolic changes in each subpathway, we used differential abundance analysis[45] for three comparisons (MYCN

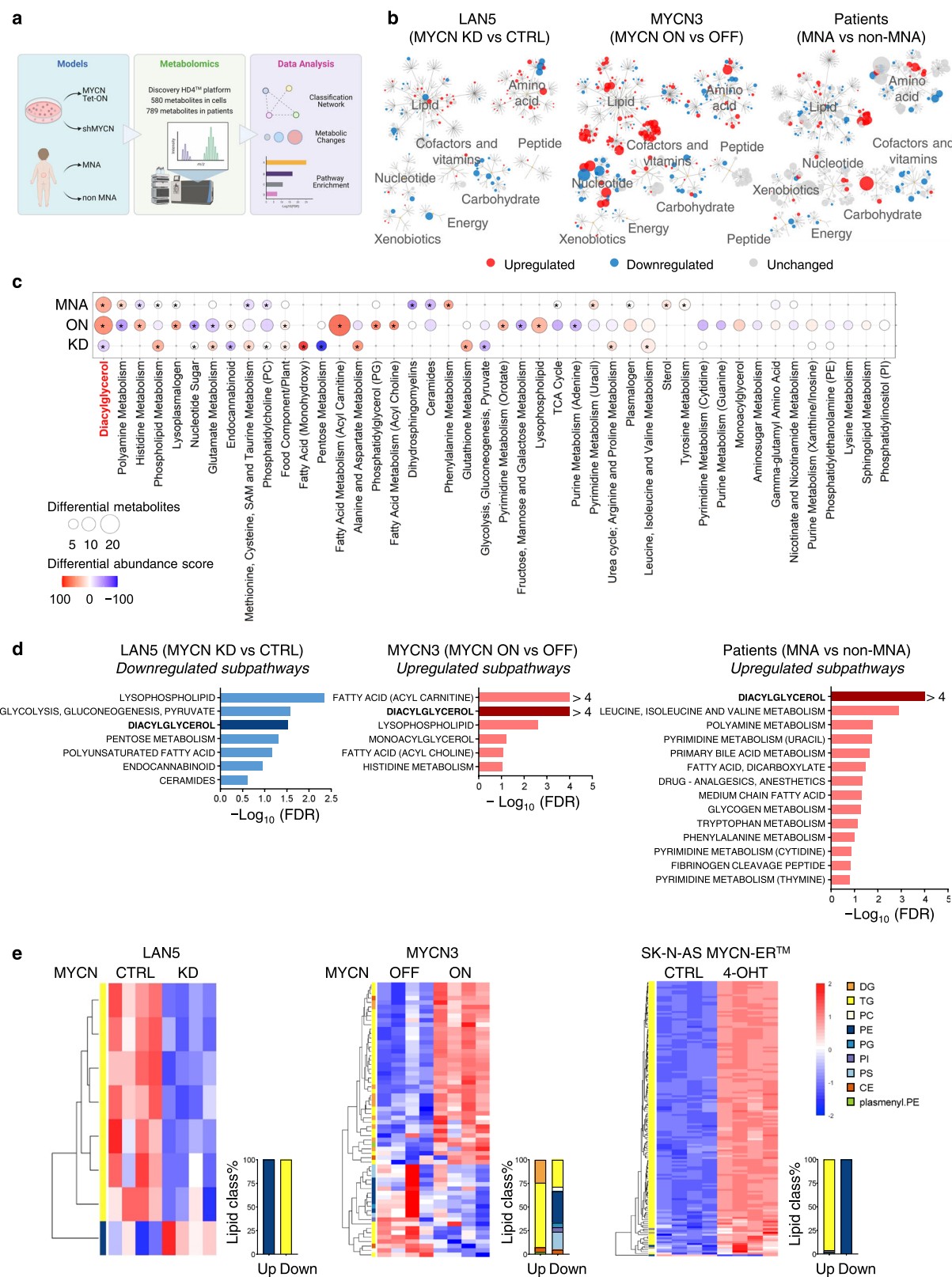

KD vs. CTRL, MYCN-ON vs. MYCN-OFF, and MNA vs. non-MNA). An abundance score of 100 indicates that 100% of the subpathway metabolites are significantly upregulated ($p \leq 0.05$), whereas a score of −100 indicates that 100% of the subpathway metabolites are significantly downregulated ($p \leq 0.05$). Significantly enriched subpathways (FDR < 0.25, Benjamini–Hochberg

adjusted $p$-value) are marked with asterisks "*". Subpathways with the same number of "*" designations were ranked by the average absolute differential abundance score. Notably, the DG group was the most differentially abundant pathway, consistently upregulated by MYCN in all three systems (differential abundance score: MYCN KD = − 25; MYCN ON = 70; MNA =

**Fig. 1 MYCN reprograms NB metabolism. a** Untargeted metabolomics profiling workflow using UHPLC-MS/MS and GC-MS (Discovery HD4™ platform, Metabolon Inc.) in LAN5 cells (MYCN KD for 0, 72, and 96 h, $n = 4$ each), MYCN3 cells (MYCN-ON for 0, 48, and 72 h, $n = 4$ each), and primary tumors (MNA, $n = 18$; non-MNA, $n = 18$). **b** Metabolite classification network. Each circle represents a metabolite. Circle size indicates absolute $\log_2FC$ of metabolite level in comparisons of MYCN KD 72 h vs. CTRL, MYCN-ON 72 h vs. MYCN-OFF and MNA vs. non-MNA. Red=upregulated metabolite; blue=downregulated metabolite ($p \leq 0.05$). One-way ANOVA or Welch's two-sample $t$-test was used to compare metabolite levels between groups. **c** Metabolic changes within subpathways. Comparison groups are the same as in (**b**). Differential abundance scores were calculated from subpathways containing at least three measured metabolites. 100 or $-100 =$ all metabolites in the subpathway are upregulated or downregulated ($p \leq 0.05$). Circle size=number of differentially altered metabolites ($p \leq 0.05$). *indicates FDR < 0.25, hypergeometric test with $p$-value adjusted by Benjamini–Hochberg procedure. **d** Pathway enrichment analysis using GSEA-based algorithm. Subpathways with FDR < 0.25 were selected and ranked by $-\log_{10}$(FDR). Red=upregulated subpathway; blue=downregulated subpathway. **e** Lipidomics profiling in LAN5 shMYCN (CTRL and MYCN KD for 72 h), MYCN3 [MYCN-OFF (-DOX) and MYCN-ON (+DOX) for 72 h] and SK-N-AS MYCN-ER™ [MYCN-OFF (−4-OHT) and MYCN-ON (+4-OHT) for 48 h] cells ($n = 4$ each). Two-sided unpaired $t$-test; $p$-value adjusted by Benjamini–Hochberg procedure to obtain FDR. Significantly altered lipids (FDR < 0.25) were selected for heatmap (color is scaled by Z-score: red=upregulated; blue=downregulated). Percentages of upregulated and downregulated lipid classes are shown in stacked bar graphs. DG diacylglycerol, TG triacylglycerol, PC phosphatidylcholine, PE phosphatidylethanolamine, PG phosphatidylglycerol, PI phosphatidylinositol, PS phosphatidylserine, CE cholesteryl ester, plasmenyl. PE plasmenyl phosphatidylethanolamine. KD knockdown, MNA *MYCN*-amplified, non-MNA non *MYCN*-amplified. Source data are provided in the Source Data file.

56; Fig. 1c). Other MYCN-altered subpathways included: phospholipid (PL), glutamate, endocannabinoid, methionine, cysteine, *S*-adenosylmethionine (SAM) and taurine metabolism, and phosphatidylcholine (PC) (FDR < 0.25, Fig. 1c and Supplementary Fig. 1b). To further support our findings, we performed gene set enrichment analysis (GSEA; Fig. 1d and Supplementary Data 2). DG was the most enriched subpathway in MNA patients compared with non-MNA patients and was consistently induced by MYCN in both in vitro systems (FDR < 0.25, Fig. 1d). Polyamine metabolism was also upregulated in MNA patients, consistent with previous studies showing that MYCN activates polyamine metabolism[26]. The fatty acyl carnitine group was highly enriched upon MYCN activation, suggesting the upregulation of FA oxidation, but no enrichment was observed in other systems. Collectively, these results suggest that MYCN alters lipid metabolism and promotes DGs accumulation in NB tumors.

To specifically evaluate the lipid metabolic changes induced by MYCN, we performed targeted lipidomics in LAN5 shMYCN, MYCN3 Tet-On cells, and SK-N-AS MYCN-ER™ cells, in which MYCN-mediated transcription is conditionally activated by 4-hydroxytamoxifen (4-OHT)[23]. Significantly altered lipids are represented in the heatmap (FDR < 0.25, Fig. 1e and Supplementary Data 3). All TGs (downstream of DGs) were significantly downregulated upon genetic depletion of MYCN (FDR < 0.25, Fig. 1e, left), whereas most DGs and TGs were upregulated upon MYCN induction (FDR < 0.25, Fig. 1e, middle and right). Because FAs are required for glycerolipid synthesis, we analyzed the FA chain compositions of MYCN-upregulated glycerolipids across the three systems. The FA chains 14:0, 16:0, 16:1, and 18:1 were the most represented (>30%, Supplementary Fig. 1c), suggesting that these FAs contribute to glycerolipid synthesis. To examine how MYCN alters FA compositions, we applied mass spectrometry to profile FAs in MYCN3 Tet-On cells and TH-MYCN$^{+/+}$ transgenic mice, in which neural crest-specific MYCN expression drives spontaneous NB formation recapitulating human disease[3]. NB cells and tumors expressing high MYCN levels also contained high levels of 14:0, 16:1, or 18:1 (FDR < 0.25, Supplementary Fig. 1d–e), suggesting that MYCN upregulates these FAs for glycerolipid synthesis. In addition, MYCN increased the ratios of 14:1 to 14:0 and 16:1 to 16:0 (Supplementary Fig. 1d–e), thus MYCN also promotes FA desaturation. Collectively, these results indicate that MYCN changes the abundance of FAs required for glycerolipid accumulation.

**MNA cell survival relies highly on FA uptake.** FAs can be synthesized in cells or imported from the microenvironment. To dynamically examine how MYCN regulates FA synthesis, we added deuterated water ($D_2O$) and $[U-^{13}C]16:0$ to SK-N-AS MYCN-ER™ cell culture medium to trace de novo FA synthesis and desaturation, respectively. MYCN activation (4-OHT for 24 h) promoted de novo FA synthesis by increasing deuterium-labeled 16:0, and enhanced FA desaturation by increasing the ratios of $[^{13}C_{16}]16:1$ to $[^{13}C_{16}]16:0$ and $[^{13}C_{16}]18:1$ to $[^{13}C_{16}]18:0$ ($p < 0.05$, Fig. 2a). To determine the effects of MYCN on FA uptake, we traced changes in cell media supplemented with $[^{13}C_{16}]16:0$, the most abundant FA in mammalian cells, as an indicator of uptake. MYCN activation significantly enhanced FA uptake ($p < 0.05$, Fig. 2a), suggesting that MYCN promotes lipid accumulation by enhancing both FA synthesis and uptake. These findings were validated in a second MYCN system (MYCN3 Tet-On cells, $p < 0.05$, Supplementary Fig. 2a), indicating that MYCN enhances intracellular FA availability for lipid synthesis.

To assess the impact of FA synthesis and uptake on cell survival, we evaluated NB cell viability [MNA: LAN5, IMR32, SK-N-BE(2c); non-MNA: SH-SY5Y, SHEP, SK-N-AS] and normal cells (ARPE-19, C2C12, and HS-5) after treatment with two FA synthesis inhibitors (A939572, an SCD1 inhibitor and orlistat, an FASN inhibitor)[22,34] or two FA uptake inhibitors (CB16.2 and CB5, which target FATP2)[46] (Fig. 2b). NB cells were more sensitive to FA uptake inhibition (MNA IC$_{50}$: 0.5–7.9 µM; non-MNA IC$_{50}$: 1.4–10.4 µM) than to FA synthesis inhibition (MNA IC$_{50}$: 7.6–60.7 µM; non-MNA IC$_{50}$: 9.2–77.5 µM), suggesting that NB cells actively use exogenous FA for survival. CB16.2 was toxic to all tested normal cells, despite having the highest efficacy in NB cells. Conversely, CB5 was highly effective in NB cells and did not elicit cytotoxicity against normal cells. MYCN enhances SCD1 activity (Supplementary Figs. 1d–e and 2a), whereas A939572 suppresses SCD1 activity ($p < 0.0001$) and de novo FA synthesis ($p < 0.05$) in MNA cells (Supplementary Fig. 2b, left). However, SCD1 inhibition did not effectively inhibit cell growth and stimulated compensatory dose-dependent FA import from the media ($p < 0.05$; Supplementary Fig. 2b, right), suggesting that exogenous FA uptake may reduce cell sensitivity to FA synthesis inhibition. To test this hypothesis, we evaluated the viability of MNA cells in complete and delipidized media with and without A939572. The removal of exogenous lipids significantly enhanced the cytotoxicity of A939572 (IC$_{50}$ from 41.0 µM to 2.2 µM), which was partially rescued by FAs supplementation (Supplementary Fig. 2c). Pharmacological inhibition of FA uptake via CB5 also enhanced the cytotoxic effects of A939572 (IC$_{50}$ from 39.3 µM to 0.4 µM) and increased cell apoptosis ($p < 0.05$, Supplementary Fig. 2d). These results suggest that NB cells import exogenous FAs as a compensatory mechanism to evade FA synthesis inhibition.

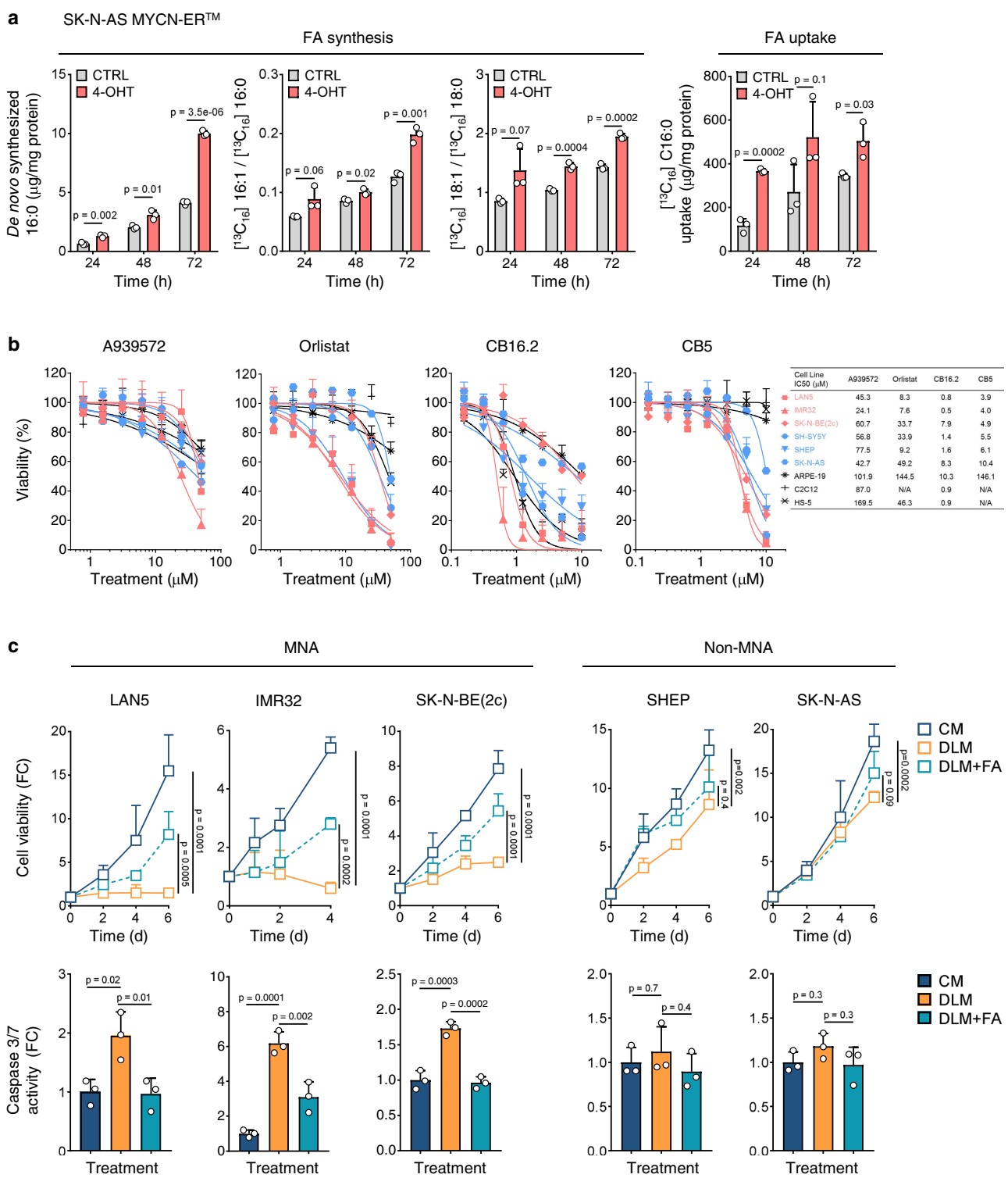

**Fig. 2 MNA cell survival relies on FA uptake. a** Stable isotope tracing of FA synthesis and FA uptake in SK-N-AS MYCN-ER™ cells with or without 4-OHT (500 nM) for 24, 48, and 72 h. Mean ± SD ($n = 3$); two-sided unpaired $t$-test per time point. **b** Viability of NB and normal cells upon 72 h treatment with FA synthesis inhibitors (A939572, orlistat) and FA uptake inhibitors (CB16.2 and CB5). IC$_{50}$ calculated in GraphPad Prism (7.01). Mean ± SD ($n = 3$). **c** Top, cell viability in complete media, delipidized media, and delipidized media supplemented with 0.025% FAs. MNA: LAN5 (0–6 day), IMR32 (0–4 day) and SK-N-BE(2c) (0–6 day); non-MNA: SHEP (0–6 day) and SK-N-AS (0–6 day). Mean ± SD ($n = 3$); two-way ANOVA with Dunnett's multiple comparisons test. Bottom, Caspase 3/7 activity in complete media, delipidized media, and delipidized media supplemented with 0.025% FAs for 4 days. Mean ± SD ($n = 3$); one-way ANOVA with Dunnett's multiple comparisons test. FC fold change, CM complete media, DLM delipidized media. Source data are provided in the Source Data file.

To determine whether MYCN-driven FA uptake supports cell survival, we examined the viability of three MNA (LAN5, IMR32, and SK-N-BE(2c)) and two non-MNA (SHEP and SK-N-AS) cell lines and SK-N-AS MYCN-ER™ cells with and without MYCN activation under complete and delipidized media conditions. Deprivation of exogenous lipids reduced the viability of MNA and MYCN-ER activated cells (+4-OHT) to a greater extent than non-MNA and MYCN-ER CTRL cells (−4-OHT) (Fig. 2c and Supplementary Fig. 2e). Moreover, FA supplementation partially restored viability of MNA and MYCN-ER activated cells ($p < 0.05$; Fig. 2c and Supplementary Fig. 2e), suggesting that exogenous FAs support MYCN-driven cell survival. Similarly, lipid deprivation induced cell apoptosis in MNA and MYCN-ER-activated cells (Fig. 2c and Supplementary Fig. 2e), which was significantly alleviated by FA supplementation ($p < 0.05$), suggesting that exogenous FAs selectively promote cell survival in MYCN-driven cells. Collectively, our data indicate that MYCN-driven cell growth depends on exogenous FAs.

**The FA transporter *SLC27A2* is a direct target of MYCN.** To further elucidate how MYCN regulates FA import, we assessed the expression of membrane FA transporter genes (*SLC27A1–6* and *CD36*) in multiple MYCN models. *SLC27A2* was remarkably upregulated in SK-N-AS MYCN-ER™ cells upon MYCN activation (10 fold at 48 h, $p = 0.0001$, Fig. 3a) and in MYCN3 cells upon MYCN induction ($p = 0.006$, Supplementary Fig. 3a). Moreover, *SLC27A2* was the only significantly downregulated transporter when MYCN was turned off in Tet-21/N cells ($p = 0.01$, Fig. 3b), suggesting that *SLC27A2* is selectively regulated by MYCN.

To assess MYCN binding to promoter regions of membrane FA transporters (*SLC27A1–6* and *CD36*), we performed MYCN ChIP-qPCR analysis in both MNA cells and MYCN Tet-Off cells (Tet-21/N). We found significant enrichment (>15 fold) of MYCN binding to the promoter region of *SLC27A2* (chr15 [hg38]: 50,182,222-50,182,302), which contains a non-canonical E-box (CACCTG) in MNA cells (LAN5 and IMR32, Supplementary Fig. 3b–c and Supplementary Data 4). Moreover, turning off MYCN almost completely abrogated MYCN binding (Fig. 3c). We then asked whether c-MYC could also bind to *SLC27A2*. c-MYC ChIP-qPCR analysis in SH-SY5Y NB cells with high c-MYC expression showed that c-MYC binds to the same promoter region but with lower affinity (4 fold enrichment; Supplementary Fig. 3b–c). To determine whether MYCN promoter binding results in gene transcription, we fused a downstream luciferase gene to the promoters of *SLC27A1*, wild-type and mutant *SLC27A2*, and *ODC1* (a known MYCN target)[47] and evaluated luciferase activity. Turning off MYCN significantly reduced wild-type *SLC27A2*-fused luciferase activity ($p = 0.01$, Supplementary Fig. 3d). However, mutation of the non-canonical E-box (CACCTG to GAATTC) in the *SLC27A2* promoter abrogated this effect (Supplementary Fig. 3d), suggesting that MYCN activates *SLC27A2* transcription via binding to this region. Although MYCN also binds to the *SLC27A1* promoter (Fig. 3c and Supplementary Fig. 3b), no changes in *SLC27A1* transcription activity were detected when MYCN was turned off (Supplementary Fig. 3d). Altogether, our study identified *SLC27A2* as a direct transcriptional target of MYCN.

Because MYCN selectively upregulates *SLC27A2* expression, we next asked whether *SLC27A2* expression correlates with *MYCN* expression, activity, or amplification status in NB patients. *SLC27A2* expression positively correlated with *MYCN* expression and activity, as indicated by the summed expression score for 157 MYCN target genes[48] ($p < 0.05$, Fig. 3d) in a large patient cohort (cohort 1: GSE45547, $n = 649$). By contrast, mRNA levels of other FA transporters did not correlate with *MYCN* expression or

activity, although *CD36* expression positively correlated with c-*MYC* expression ($p < 0.05$, Fig. 3d, left). High expression of *SLC27A2* also correlated with *MYCN* amplification ($p = 8.1e^{-45}$) and stage 4 disease ($p < 0.01$) (Fig. 3d), and strongly predicted poor overall (OS) and event-free survival (EFS) in patients of all stages (OS: $p = 1.0e^{-8}$; EFS: $p = 4.7e^{-6}$) and stage 3–4 high-risk patients (OS: $p = 2.7e^{-4}$; EFS: $p = 4.2e^{-2}$, Fig. 3e). Compared with other long-chain FA transport-associated genes (GO: 0015909, $n = 75$), *SLC27A2* ranked top in predicting poor clinical outcomes (R2, Fig. 3f, red dots). These findings were validated in a second patient cohort (cohort 2: GSE85047, $n = 283$, Supplementary Fig. 3e–g), suggesting that *SLC27A2* is a critical transporter for NB survival. To explore potential for targeting *SLC27A2*, we compared *SLC27A2* mRNA expression in three NB cohorts (Lastowska, GSE13136, $n = 30$; Hiyama, GSE16237, $n = 51$; Versteeg, GSE16476, $n = 88$) with normal tissue cohort (Adrenal Gland, SN_ADGL, $n = 13$; Neural Crest, GSE14340, $n = 5$; Normal Various, GSE7307, $n = 504$ including 108 types of normal tissues; datasets generated from the same u133p2 platform and normalized by MAS5.0). Median *SLC27A2* expression was higher in MNA patients than in normal tissues (Supplementary Fig. 3h), suggesting that *SLC27A2* could function as a therapeutic target in NB.

**Genetic and pharmacological inhibition of *SLC27A2* impairs NB survival.** To determine the impact of *SLC27A2* expression on cell survival, we analyzed survival outcomes in 789 cell lines after *SLC27A2* knockout using the CRISPR (Avana) Public 20Q4V2 dataset (Broad Institute, USA)[49]. The results were grouped according to primary disease and ranked using a dependency score. NB ranked tenth among 30 primary diseases (dependency score = −0.23), suggesting that NB cells depend on *SLC27A2* for survival (Supplementary Fig. 4a).

To confirm whether *SLC27A2* is required for NB survival, we genetically depleted *SLC27A2* via shRNA-mediated gene silencing (two sequences) in MNA LAN5 cells (Fig. 4a), which express high basal *SLC27A2* levels (Supplementary Fig. 4b). *SLC27A2* KD in LAN5 cells effectively reduced FA uptake ($p < 0.01$, Fig. 4b) and impaired cell growth ($p < 0.0001$, Fig. 4c) and colony forming capacity ($p < 0.05$, Fig. 4d), suggesting that *SLC27A2* is required for cell growth. This phenotypic outcome was confirmed in a second MNA cell line (IMR32) (Supplementary Fig. 4c) and was selective to MNA cells, as *SLC27A2* depletion in non-MNA cells (SK-N-AS) did not affect FA uptake or cell growth (Supplementary Fig. 4c). To further examine whether *SLC27A2* is required for in vivo tumor growth, we orthotopically implanted MNA LAN5 shCTRL and sh*SLC27A2* cells into NCr nude mice. Silencing *SLC27A2* significantly reduced tumor growth (assessed by MRI imaging, $p = 0.048$) and tumor weights ($p = 0.01$, Fig. 4e–f). These effects were associated to a significant reduction of intratumoral neutral lipids (Oil Red O staining, $p = 0.003$, Fig. 4g). Furthermore, depletion of *SLC27A2* reduced tumor proliferation (Ki67 staining, $p = 0.004$, Supplementary Fig. 4d) and increased tumor apoptosis (cleaved Caspase-3 staining, $p < 0.0001$, Supplementary Fig. 4e) without altering MYCN expression ($p = 0.4$, Supplementary Fig. 4f). Lipidomic profiling of shCTRL ($n = 8$) and sh*SLC27A2* ($n = 8$) tumors demonstrated downregulation of the majority of DGs and TGs upon *SLC27A2* KD (FDR < 0.25, absolute FC > 2, Fig. 4h and Supplementary Data 3), confirming that inhibition *of SLC27A2* effectively reverts MYCN-induced glycerolipid accumulation. Collectively, these data indicate that *SLC27A2* is required for NB cell survival and tumor growth.

To pharmacologically target FATP2 (encoded by *SLC27A2*), we used the small-molecule FATP2 inhibitor CB5[46], which elicits

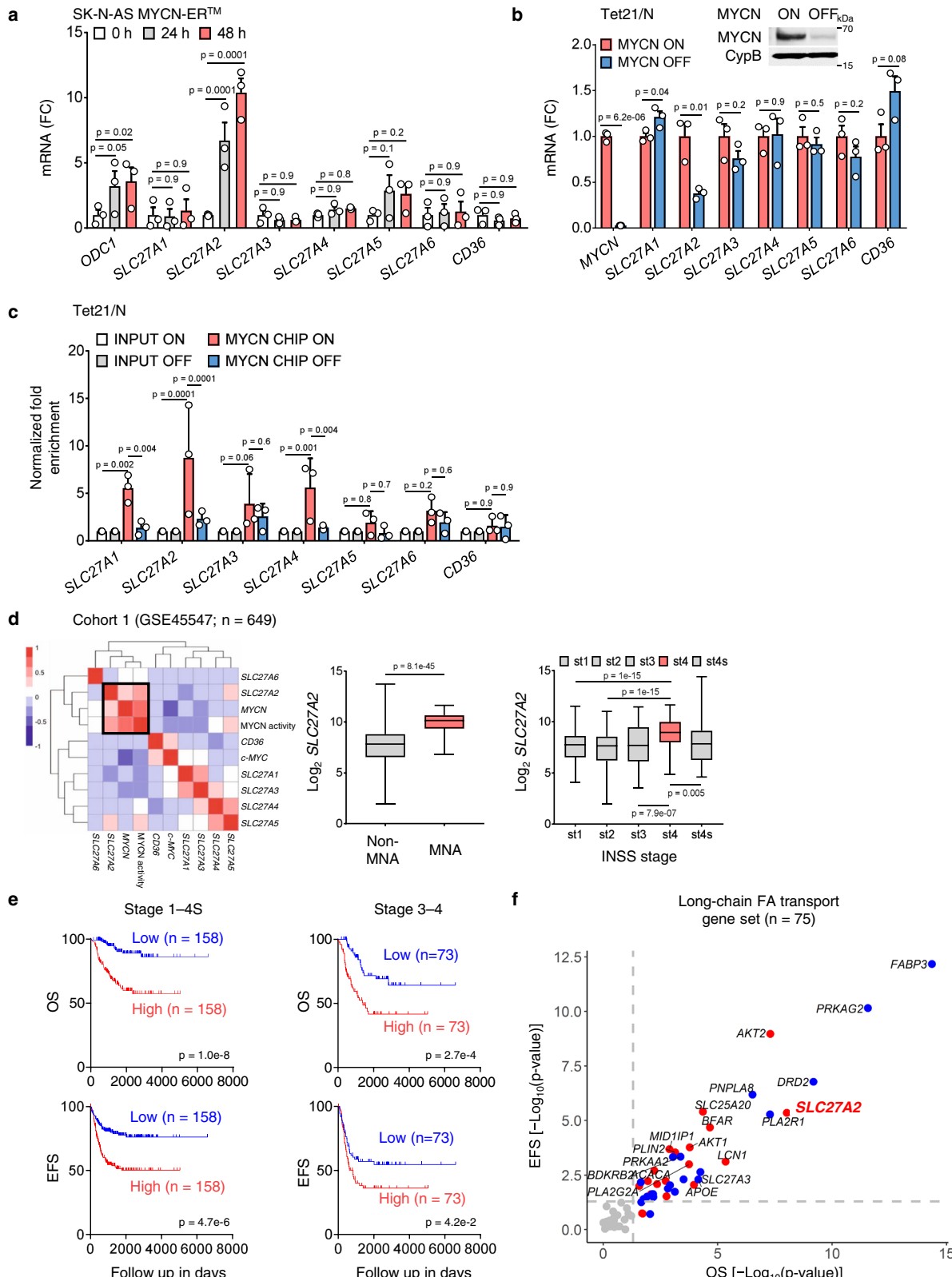

selective cytotoxicity against NB cells but spares normal cells (Fig. 2b). We first validated the ability of CB5 to block FA uptake in NB cells by both stable isotope tracing (Supplementary Fig. 2d) and staining with a fluorescent-labeled FA analog BODIPY™ 500/510 C1, C12. CB5 preferentially blocked FA uptake in MNA LAN5 cells ($p < 0.05$) compared with non-MNA SHEP cells (Fig. 4i). Because the FATP2 inhibitor CB16.2 also suppresses

FATP1 activity[41], we verified the specificity of CB5 for FATP2. Ectopic overexpression of both FATP1 and FATP2 increased FA uptake in MNA cells. However, CB5 only reduced FA uptake in FATP2-overexpressing cells (Supplementary Fig. 4g), suggesting that CB5 specifically targets FATP2 in NB. CB5 more effectively inhibited cell viability and induced apoptosis (determined by Caspase 3/7 activity) in MNA cells than in non-MNA cells,

**Fig. 3 MYCN directly upregulates FA transporter *SLC27A2*. a** FA transporters (*SLC27A1–6*, *CD36*) and *ODC1* mRNA expression in SK-N-AS MYCN-ER™ cells (1 μM 4-OHT for 0, 24, 48 h). Mean±SEM (*n* = 3); two-way ANOVA with Dunnett's multiple comparisons test. **b** FA transporters (*SLC27A1–6*, *CD36*) mRNA expression in Tet21/N cells (±2 μg/mL DOX for 24 h). Mean±SEM (*n* = 3); two-sided unpaired *t*-test. MYCN protein expression in Tet21/N cells (−DOX, MYCN-ON and +DOX, MYCN-OFF). **c** MYCN ChIP-qPCR assays in TET21/N cells (±2 μg/mL DOX for 48 h). Input and MYCN ChIP samples were analyzed by qPCR using specific primers for *SLC27A1–6* and *CD36*. Mean±SD (*n* = 3); two-way ANOVA with Dunnett's multiple comparisons test. **d** Gene expression analysis in Cohort 1 (GSE45547). Left, correlation matrix of transporter gene expression, *MYCN* expression/activity, and *c-MYC* expression. Correlations with *p*-values < 0.05 are represented in the heatmap. Red = positive; blue = negative correlation. Middle, *SLC27A2* expression in MNA (*n* = 93) and non-MNA patients (*n* = 550). Two-sided unpaired Welch's *t*-test. Right, *SLC27A2* expression in stage 1–4 S patients (stage 1: *n* = 153; stage 2: *n* = 113; stage 3: *n* = 91; stage 4: *n* = 214; stage 4 s: *n* = 78). One-way ANOVA with Tukey's multiple comparisons test. Box plots indicate median (middle line), 25th and 75th percentiles (box), as well as min and max (whisker). **e, f** Survival analyses in Cohort 1 (GSE45547). **e** OS and EFS rates for stage 1–4 S and stage 3–4 patients with high (top third) or low (bottom third) *SLC27A2* expression. **f** OS and EFS predictions for genes in the long-chain FA transport geneset (GO: 0015909); –log$_{10}$(*p*-value). Kaplan–Meier method was used to plot survival curves, and log-rank test was used for statistical analysis. Red=high expression has poor prognosis (*p* < 0.05); blue = low expression has poor prognosis (*p* < 0.05); gray = no significance. FC fold change, OS overall survival, EFS event-free survival. Source data are provided in the Source Data file.

without affecting normal cells (Figs. 2b and 4j), suggesting that targeting FATP2 selectively impairs MNA cell survival. The selectivity of CB5 was further supported by the preferential induction of cleaved PARP and cleaved Caspase-3 expression (markers of apoptosis) in MNA LAN5 and IMR32 cells compared with non-MNA SHEP cells (*p* < 0.05, Fig. 4k and Supplementary Fig. 5a). CB5 also inhibited MYCN but not c-MYC protein expression, supporting the selective targeting of MNA cells. p53 and its downstream target p21(Waf1/Cip1) play critical roles in cell cycle, proliferation, and apoptosis in the setting of *MYCN* amplification. Although p53 can be directly upregulated by MYCN as a mechanism for MYCN-induced apoptosis[50], we found that CB5 increased both p53 and p21(Waf1/Cip1) protein expression in MNA cells (*p* < 0.05, Fig. 4k and Supplementary Fig. 5a). Thus, CB5 inhibits MYCN and activates p53 signaling to suppress cell growth and promote apoptosis.

**Targeting FA transport effectively suppresses NB tumor growth.** To determine the contribution of FA transport to tumor growth, we evaluated the anti-tumor activity of CB5 in multiple preclinical NB models. We orthotopically implanted MNA LAN5 luciferase-expressing cells and non-MNA SK-N-AS cells into the renal capsule of NCr nude mice (Fig. 5a). After tumor engraftment, mice were randomly assigned to CTRL (vehicle) or CB5 (25 mg/kg, twice a day [*b.i.d.*], intraperitoneal injection (*i.p.*) for 2 weeks) treatment groups. CB5 significantly inhibited the growth of LAN5 xenografts as measured by luciferase activity (*p* = 0.002) and tumor weights (*p* = 0.03, Fig. 5b and Supplementary Fig. 5b). However, CB5 did not reduce tumor volumes and weights of SK-N-AS xenografts (Fig. 5c and Supplementary Fig. 5c), suggesting that CB5 preferentially inhibits MNA tumors. Notably, CB5 treatment did not cause toxicity assessed by mouse general clinical conditions and weight changes during treatment (Supplementary Fig. 5b–c).

The TH-MYCN transgenic model is an aggressive MYCN-induced de novo NB model[3]. To assess the anti-cancer activity of CB5, we generated an orthotopic allograft model of NB by implanting a TH-MYCN$^{+/+}$ tumor into NCr nude mice (Fig. 5d). Tumors developed after 2 weeks, at which time mice were randomly assigned to CTRL (vehicle) or CB5 (25 mg/kg, *b.i.d.*) treatment groups. MRI was performed on treatment days 1 and 14 to monitor tumor growth. CB5 significantly reduced tumor volumes (*p* = 0.006, Fig. 5e and Supplementary Fig. 5d) and weights (*p* = 0.01, Fig. 5f) in this model, and no signs of toxicity were observed during the study (Supplementary Fig. 5d). Moreover, mice responsive to CB5 treatment showed lower neutral lipid levels than CTRL mice (*p* = 0.004, Fig. 5g), suggesting that CB5 blocks lipid accumulation in vivo. This is likely due to the CB5-mediated inhibition of MYCN and MYCN-targeted FA

synthesis and transport protein expression (SCD1, ACC, and FATP2; Supplementary Fig. 5d). Two tumors escaped CB5 treatment (Fig. 5f). Supporting our findings, these tumors did not exhibit lipid inhibition and showed signs of MYCN activation and FA synthesis/transport activity (SCD1 and FATP2, Supplementary Fig. 5d).

One caveat to using nude mice as preclinical models is that they lack an immune microenvironment. To evaluate the efficacy of blocking FA transport in the presence of an intact immune microenvironment, we generated a TH-MYCN$^{+/+}$-derived orthotopic syngeneic mouse model (Fig. 5h) by implanting a TH-MYCN$^{+/+}$ tumor into the renal capsule of wild-type immunocompetent 129×1/svj mice. After 2 weeks, mice were treated with either CTRL (vehicle) or CB5 (30 mg/kg, *b.i.d. i.p.*) for 2 weeks. CB5 treatment blocked tumor growth (Fig. 5h, *p* < 0.0001) without apparent toxicity (Supplementary Fig. 5e) in this model. To evaluate the long-term effects of blocking FA uptake in MNA NBs, we used a patient-derived orthotopic xenograft model (Fig. 5i). Cells prepared from a primary MNA stage 4 NB tumor (P0) were implanted into the renal capsule of NOG mice (P1). The tumor was then passaged in NOG mice to P4 and in NCr nude mice to P6. In this model, tumors initiate approximately 5 weeks after implantation and mice succumb to disease burden 8–10 weeks after implantation. We asked whether blocking FA uptake through long-term CB5 treatment could prevent MNA tumor development and prolong animal survival. CB5 treatment was applied 2 weeks after implantation when tumors had not initiated. Mice received CTRL (vehicle) or CB5 (25 mg/kg, *b.i.d.*, *i.p.*) for 6 weeks and tumor growth was monitored by MRI. CB5 did not prevent tumor initiation (Fig. 5i). However, chronic CB5 treatment significantly prolonged animal survival (*p* = 0.004, Fig. 5i) without notable toxicity (Supplementary Fig. 5f), suggesting that blocking FA uptake can suppress primary MNA tumor growth and extend survival.

**Targeting FA transport sensitizes NB to conventional chemotherapies.** Elevated lipid metabolism promotes acquired resistance to chemotherapy. Targeting FA synthesis (FASN), oxidation (CPT1), or uptake (CD36) sensitizes cancer cells to chemotherapy in adult models[40,51,52]. Thus, we asked whether FATP2 inhibition enhanced the anti-tumor activity of conventional chemotherapies, such as etoposide (VP16, a topoisomerase II inhibitor) or temozolomide (TMZ, a DNA alkylating agent), which are used as induction or relapse therapies. We first evaluated the synergy between CB5 and either VP16 or TMZ by assessing cell viability under single and combination treatment conditions, assigning a synergy score using the Bliss model (>10 indicates synergy)[53]. CB5 synergizes with both VP16 and TMZ in MNA cells (Fig. 6a and Supplementary Fig. 6a–b). In addition,

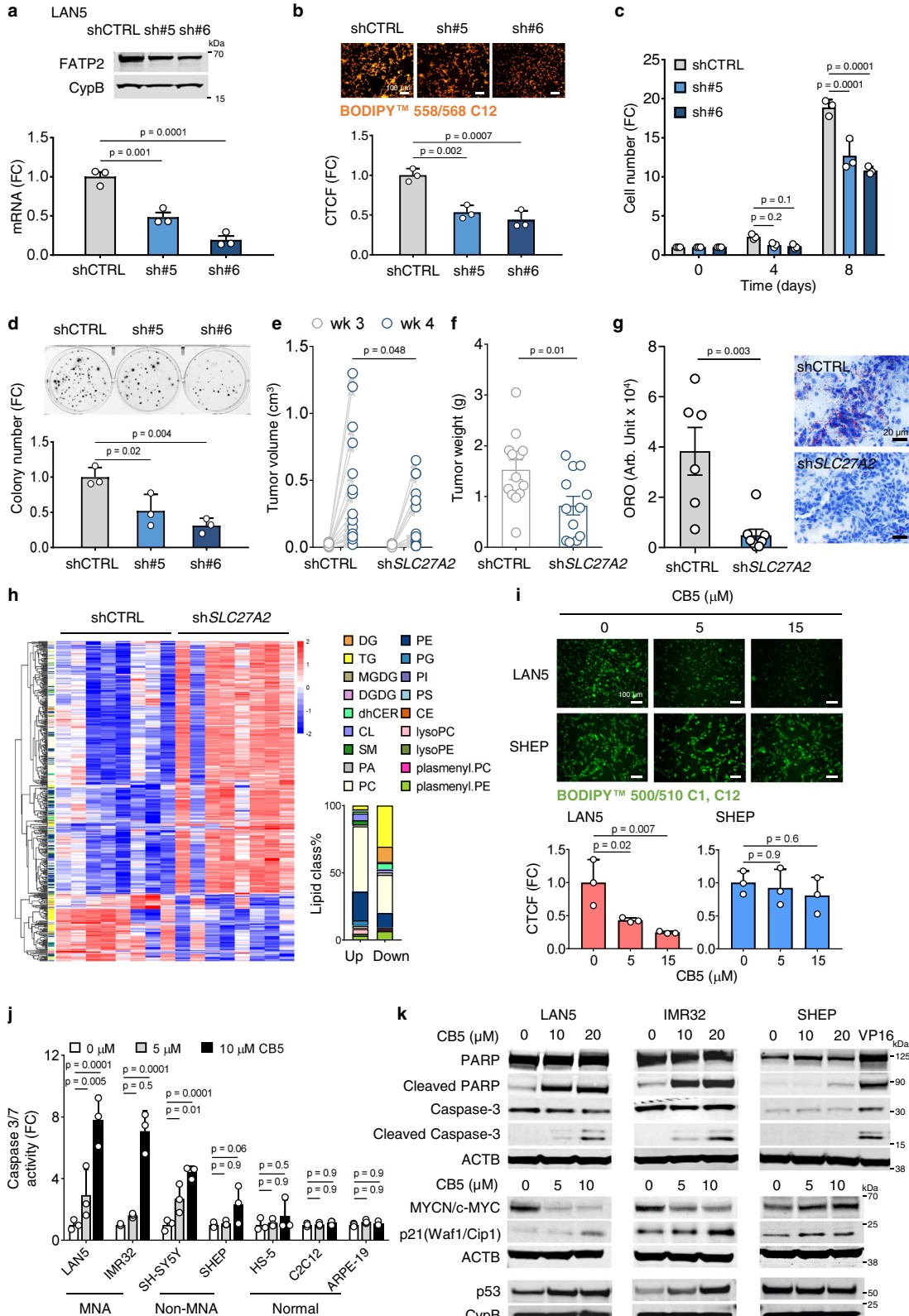

both CB5 + VP16 and CB5 + TMZ combinations increased Caspase-mediated apoptosis compared with single-drug treatments in MNA cells ($p < 0.05$, Fig. 6b and Supplementary Fig. 6c), suggesting that these combination therapies are highly effective in the setting of *MYCN* amplification.

Because the CB5 + VP16 combination showed promising in vitro synergistic effects, we assessed the anti-tumor activity of this combination in MNA LAN5-derived orthotopic xenografts (Fig. 6c). Combination therapy (CB5: 15 mg/kg, *b.i.d.*, *i.p.* 6 days/ week and VP16: 8 mg/kg, *i.p.* 3 days/week) markedly reduced final tumor weights compared with monotherapies ($p < 0.0005$, Fig. 6c). Moreover, CB5 + VP16 demonstrated enhanced effects on tumor cell proliferation and apoptosis ($p \leq 0.05$, Supplementary Fig. 6d–e) without further reducing MYCN expression

**Fig. 4 Suppressing FA uptake impairs NB cell survival. a** Silencing *SLC27A2* in LAN5 cells. Two sh*SLC27A2* GIPZ vectors tested, with empty GIPZ vector as control. Mean ± SEM ($n = 3$). **b** FA uptake in LAN5 shCTRL and sh*SLC27A2* cells. Cells stained with the FA analog BODIPY™ 558/568 C12 and quantified as CTCF by ImageJ2. Mean±SD ($n = 3$). **c** Cell growth in LAN5 shCTRL and sh*SLC27A2* cells. Mean ± SD ($n = 3$). **d** Clonogenic assay in LAN5 shCTRL and sh*SLC27A2* cells. Mean ± SD ($n = 3$); **e–h** LAN5 shCTRL and sh*SLC27A2* orthotopic xenograft model. **e** Tumor volumes at weeks 3 and 4 post-implantation. CTRL = 15; sh*SLC27A2* = 14; two-way ANOVA with Sidak's multiple comparisons test. **f** Tumor weights at week 5 post-implantation. Mean ± SEM (CTRL = 13, sh*SLC27A2* = 12); two-sided unpaired Mann–Whitney test. **g** Oil Red O staining of intratumoral lipids. Mean ± SEM (CTRL = 6, sh*SLC27A2* = 8); two-sided unpaired Mann–Whitney test. **h** Lipidomics profiling of shCTRL and sh*SLC27A2* tumors ($n = 8$ each). Lipids (FDR < 0.25, absolute FC > 2) are shown in the heatmap (color scaled by Z-score: red = upregulated; blue = downregulated). MGDG monogalactosyldiacylglycerol, DGDG digalactosyldiacylglycerol, dhCER dihydroceramides, CL cardiolipin, SM sphingomyelin, PA phosphatidic acid, lyso.PC lysophosphatidylcholine, lyso.PE lysophosphatidylethanolamine, plasmenyl.PC plasmenylphosphatidylcholine, all other abbreviations consistent with Fig. 1e. **i** FA uptake following CB5 treatment in LAN5 and SHEP cells (0–15 µM, 5 min). Cells stained with BODIPY™ 500/510 C1, C12 and quantified as CTCF by ImageJ2. Mean ± SD ($n = 3$). **j** Caspase 3/7 activity of NB and normal cells after CB5 treatment (0–10 µM, 24 h). Mean ± SD ($n = 3$). **k** Apoptosis, p53/p21, and MYCN/c-MYC protein expression in LAN5, IMR32, and SHEP with CB5 (0–20 µM, 16–24 h). VP16 (10 µM, 24 h) = positive CTRL. Representative blots from three independent experiments are shown. FC fold change, Arb. Unit arbitrary unit. **a**, **b**, and **i**, one-way ANOVA with Dunnett's multiple comparisons test; **c** and **j**, two-way ANOVA with Dunnett's multiple comparisons test. Source data are provided in the Source Data file.

compared with CB5 alone (Supplementary Fig. 6f) and with no evidence of body weight loss or normal organ toxicity (Supplementary Fig. 6g). These data suggest that blocking FA transport effectively enhances tumor responses to VP16. To then determine the long-term effects of combination therapy on animal survival, we used our MNA patient-derived orthotopic xenograft model (Fig. 6d). Mice were subjected to single or combination therapy (CB5: 15 mg/kg, *b.i.d.*, *i.p.* 6 days/week and VP16: 6 mg/kg, *i.p.* 3 days/week) for 5 weeks. Animals treated with CB5 + VP16 combination therapy survived significantly longer than those receiving single-drug therapies ($p < 0.05$, Fig. 6d). No treatments caused significant body weight loss or clinical signs of toxicity (Supplementary Fig. 6h). Collectively, our data suggest that blocking MYCN-induced metabolic reprogramming effectively enhances the anti-tumor effects of conventional chemotherapy.

## Discussion
Using a variety of biochemical and analytical approaches, we comprehensively characterized the metabolic landscape of NB tumors and defined *MYCN* amplification as a major driver of distinct metabolic adaptations. Our metabolic screening and analyses in human tumors and cell lines across various models revealed that MYCN induces a consistent accumulation of glycerolipids. Glycerolipids function as secondary messengers that activate downstream oncogenic signaling and serve as FA reservoir for energy storage and the prevention of toxic FAs accumulation to support tumor growth[29]. *MYCN* induction or amplification resulted in a distinct glycerolipid signature mainly characterized by a robust increase in DGs. FAs can be funneled into various metabolic pathways to synthesize more complex lipid species, including DGs (which are precursors of TGs), contributing to the structural diversity of the cellular lipids. MYCN may directly regulate glycerolipid synthesis and degradation, for example, by upregulating diacylglycerol-acyltransferase 2 (DGAT2) to store excess FAs in TGs and LDs (Supplementary Fig. 7). DGAT1 and DGAT2 catalyze the esterification of acyl-CoA with DGs to form TGs[54]. Although their roles in oncogenesis remain largely unexplored, targeting DGAT1 to block FA storage induces severe oxidative stress in glioblastoma[55]. Future investigations remain necessary to elucidate how MYCN maintains lipid homeostasis and protects NB cells from oxidative damage. Here we show that targeting FA uptake effectively inhibits MYCN-induced glycerolipid accumulation and tumor growth, suggesting that FA transport is critical for these functions.

Cancer tissues show aberrant activation of de novo lipogenesis, and inhibition of enzymes within the FA biosynthesis pathway can block tumor growth[30]. FA biosynthesis contributes to cancer by providing the building blocks for biological membranes and ATP synthesis during nutrient depletion, in addition to regulating membrane trafficking and signaling pathways critical for cell survival, such as phosphatidylinositol-3,4,5-trisphosphate and sphingolipids[56,57]. MYC(N) dynamically alters de novo lipogenesis, lipid storage, and β-oxidation to generate ATP. Inhibition of MYC(N) induces LDs formation as a consequence of impaired FA oxidation[58]. We and others have also shown that MYC(N) promotes FA biosynthesis and aberrant activation of SREBP1[18] in cancer, including NB[22,23], and genetic and pharmacological interference with FA biosynthesis blocks MYC-driven tumor growth[18]. Moreover, ACC or FASN inhibition partially blocks cancer growth in a subcutaneous NB model[22]. However, the long-term efficacy and normal tissue toxicity associated with this approach remain largely unknown. We found that inhibiting FA biosynthesis via the SCD1 inhibitor A939572 triggered compensatory FA uptake, suggesting that NB cells can utilize exogenous lipids when precursors are limited or FA synthesis is impaired. We demonstrated that blocking FA uptake either through exogenous lipid deprivation or pharmacological FATP2 inhibition remarkably reduced the viability of MYCN-induced cells, promoted apoptosis, and enhanced sensitivity to FA biosynthesis inhibition. These effects could be partially rescued by FA supplementation, indicating that MYCN-driven cells depend on FA uptake for survival. The dependency on exogenous FAs is emerging in other cancers. In prostate cancer and melanoma, the membrane transporters CD36 and FATP1 respectively promote lipid accumulation and tumor progression[38,41]. Moreover, melanoma cells use FATP2 to acquire lipids from aged fibroblasts, conferring resistance to targeted therapy[42]. In NB, we found that oncogenic MYCN drives FA uptake to maintain tumor growth. By screening mRNA expression and MYCN binding to membrane FA transporters, we identified the FA transporter gene *SLC27A2* (encoding FATP2) as a direct MYCN target required for NB survival. *SLC27A2* expression in NB patients uniquely correlates with *MYCN* expression and activity. Moreover, high *SLC27A2* expression strongly predicts poor clinical outcomes when compared to other FA transporters. Collectively, these data suggest that MNA NB depends on *SLC27A2*-mediated FA uptake, making *SLC27A2* an attractive therapeutic target for high-risk disease.

*SLC27A2* is necessary for NB survival, as both genetic interference and pharmacological inhibition via the small-molecule inhibitor CB5 impaired MYCN-induced tumor growth. Supporting the selectivity of targeting *SLC27A2* in NB, normal cells were not affected by CB5 treatment, suggesting the possibility of new therapeutic opportunities. We demonstrated the anti-tumor

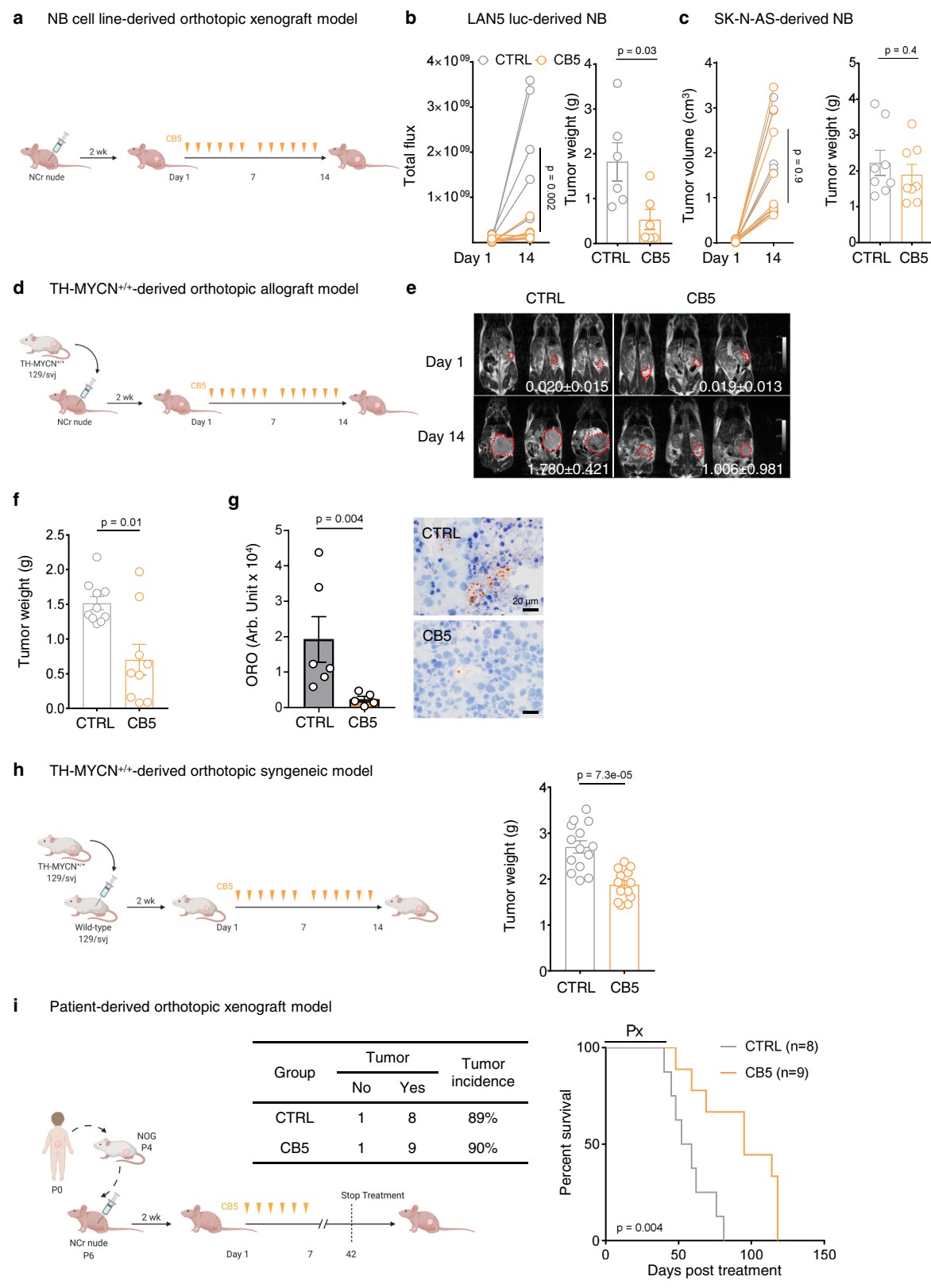

activity of CB5 in multiple MYCN-induced preclinical NB models, including cell line-derived orthotopic models, both allograft and syngeneic models, and patient-derived models, suggesting that FA uptake via *SLC27A2* represents an intrinsic vulnerability of MYCN-driven tumors that can be targeted therapeutically. *SLC27A2* is also expressed in immune cells, such as oncogenic polymorphonuclear myeloid-derived suppressor

cells, and *SLC27A2* inhibition blocks immune-suppressive activity in these cells, delaying tumor progression[59]. Future studies remain necessary to determine whether *SLC27A2* inhibition also interferes with immune cell functions and their metabolic dependencies to prohibit tumor growth.

MYCN-induced drug resistance limits the anti-tumor effects of conventional chemotherapy, and thus remains a major clinical

**Fig. 5 Suppressing FA uptake exerts anti-tumor effects in multiple preclinical models. a–c** NB cell line-derived orthotopic xenograft model. **a** LAN5 or SK-N-AS cells were orthotopically implanted in NCr nude mice. Two weeks later mice were treated with vehicle or CB5 (25 mg/kg, *b.i.d.*, 6 days/week) for 2 weeks. **b** LAN5 tumor sizes (IVIS) and weights after treatment. Mean±SEM (CTRL = 6, CB5 = 6); two-way ANOVA with Sidak's multiple comparisons test (left); two-sided unpaired Mann–Whitney test (right). **c** SK-N-AS tumor volumes (MRI) and weights after treatment. Mean ± SEM (CTRL = 8, CB5 = 8); two-way ANOVA with Sidak's multiple comparisons test (left); two-sided unpaired Mann–Whitney test (right). **d–g** TH-MYCN$^{+/+}$-derived orthotopic allograft model. **d** Cells from one TH-MYCN$^{+/+}$ tumor were orthotopically implanted in NCr nude mice. Two weeks later mice were treated with vehicle or CB5 (25 mg/kg, *b.i.d.*, 6 days/week) for 2 weeks. **e** Tumor volumes (MRI) on treatment days 1 and 14. Tumors were framed and quantified; representative images and mean ± SEM are shown (CTRL = 10, CB5 = 9). **f** Tumor weights at treatment day 14. Mean ± SEM (CTRL = 10, CB5 = 9); two-sided unpaired Mann–Whitney test. **g** Oil Red O staining of intratumoral lipids. Mean±SEM (CTRL = 6, CB5 = 5 responsive tumors); two-sided unpaired Mann–Whitney test. **h** Cells from one TH-MYCN$^{+/+}$ tumor were orthotopically implanted in syngeneic 129 × 1/svj wild-type mice. Two weeks later mice were treated with vehicle or CB5 (30 mg/kg, *b.i.d.*, 6 days/week) for 2 weeks. Tumors were weighed on treatment day 14. Mean ± SEM (CTRL = 14, CB5 = 13); two-sided unpaired Mann–Whitney test. **i** Cells from one MNA patient tumor (P6) were orthotopically implanted in NCr nude mice, and 2 weeks later mice were treated with vehicle or CB5 (25 mg/kg, *b.i.d.*, 6 days/week) for 6 weeks. Tumor incidence analyzed by Fisher's exact test (CTRL = 8, CB5 = 9). Kaplan–Meier survival analyzed by log-rank test. Arb. Unit arbitrary unit, Px treatment period. Source data are provided in the Source Data file.

obstacle[60]. Moreover, metabolic reprogramming and altered lipid metabolism also promote acquired drug resistance[27]. Increased *FASN*, *CPT1B*, and *CD36* expression correlate with poor drug response in cancer patients[40,51,52]. We found that targeting a key MYCN-dependent metabolic vulnerability, the FA uptake pathway, strongly enhanced the clinical activity of cytotoxic agents that inhibit DNA synthesis, such as VP16 and TMZ. Because FA-derived acetyl-CoA can fuel the TCA cycle to generate aspartate, the nucleotide synthesis precursor[61], we speculate that inhibiting FA uptake may limit nucleotide reservoirs for DNA synthesis. Future studies are needed to test whether FA uptake contributes to FA oxidation and nucleotide synthesis and investigate the molecular mechanisms underlying the observed synergy between CB5 and conventional chemotherapy.

In conclusion, we uncovered that MYCN promotes glycerolipid accumulation in NB. MYCN drives FA uptake by directly upregulating *SLC27A2*, which is required for glycerolipid synthesis and MYCN-induced cell survival. Genetic and pharmacological interference (via CB5) of *SLC27A2* blocks NB tumor growth. Moreover, CB5 prolongs animal survival, and synergizes with conventional chemotherapy in multiple preclinical NB models. Our study shows that MYCN-driven tumors rely heavily on FA uptake for survival, suggesting that FA uptake may represent a promising therapeutic target in high-risk MNA patients.

## Methods
**Patient sample analyses.** Frozen primary tumors (MNA, $n = 18$; non-MNA, $n = 18$) were provided by the Research Tissue Support Service (RTSS) at Texas Childrens' Hospital (TCH). Patient clinical information is summarized in Supplementary Data 1. This study was approved by the Institutional Review Board for Human Subject Research at Baylor College of Medicine and Affiliated Hospitals (BCM IRB, H-42596, H-6650). Informed consent was obtained from all participants. Compensation was not provided.

Publicly available patient data were retrieved from R2: Genomics Analysis and Visualization Platform (http://r2.amc.nl). Cohort 1 (Kocak, GSE45547, $n = 649$): data included single-color gene expression profiles from 649 NB tumors based on 44 K oligonucleotide microarrays[62]. Survival annotations are available for 479 patients. Cohort 2 (NRC, GSE85047, $n = 283$): data were profiled using the Affymetrix Human Exon 1.0 ST Array[63]. The *MYCN* activity signature was derived from a signature composed of 157 differentially expressed genes identified upon genetic depletion of *MYCN* from MNA cell lines[48]. Gene expression was then converted to a Z-score, and the *MYCN* activity score of each patient was computed by adding the Z-scores of upregulated genes and subtracting the Z-scores of downregulated genes. Pearson's correlation coefficient and *p*-value (Python scientific library) were used to analyze correlations between the expression levels of two genes or between gene expression and *MYCN* activity. Correlations with *p*-values < 0.05 are shown in the heatmap. To compare gene expression between NB and normal tissues, *SLC27A2* mRNA expression was extracted from three NB datasets (Lastowska, GSE13136, $n = 30$; Hiyama, GSE16237, $n = 51$; and Versteeg, GSE16476, $n = 88$) and three normal tissue datasets (Adrenal Gland, SN_ADGL, $n = 13$; Neural Crest, GSE14340, $n = 5$; and Normal Various, GSE7307, $n = 504$ including 108 types of normal tissues). NB and normal tissue datasets were acquired with the u133p2 platform and normalized by MAS5.0 (R2). Kaplan–Meier

analysis was used to assess how FA transporter gene expression predicts patient survival by computing OS and EFS or PFS rates of the top and bottom tertiles, stratified by expression of each gene in the long-chain FA transport gene set (GO: 0015909). *P*-values were computed by the log-rank test. OS and EFS significance [-log$_{10}$(*p*-value)] of all transporters were ranked and are shown in the scatter plot. Red dots indicate that high expression predicts poor outcome ($p < 0.05$); blue dots indicate that low expression predicts poor outcome ($p < 0.05$); gray dots indicate no significance.

**Cell culture and drugs.** The human NB cell lines IMR32 (male, RRID: CVCL_0346, CCL-127), SK-N-AS (female, RRID: CVCL_1700, CRL-2137), SH-SY5Y (female, RRID: CVCL_0019, CRL-2266) (from the American Type Culture Collection, ATCC), SHEP (female, RRID: CVCL_0524, Shohet lab, University of Massachusetts), Kelly (female, RRID: CVCL_2092, Sigma 92110411) and SK-N-BE(2c) (male, RRID: CVCL_0529, ATCC CRL-2268) (Bernardi lab, BCM), LAN5 (male, RRID: CVCL_0389, Metelitsa lab, BCM), LAN5 shMYCN (Bernardi lab), and MYCN3 (Shohet lab) were maintained in RPMI 1640 media (Lonza, NJ, USA) containing 10% FBS (Germini Bio-Products, CA, USA), 4 mM *L*-glutamine (Thermo Fisher Scientific, MA, USA), and 1% streptomycin-penicillin (Thermo Fisher Scientific). SK-N-AS MYCN-ER™ cells (Altman lab, University of Rochester, NY), as well as HS-5 bone marrow stroma cells (male, RRID: CVCL_3720, Redell lab, TCH, ATCC CRL-11882) and C2C12 mouse myoblast cells (female, RRID: CVCL_0188, Neilson lab, BCM, ATCC CRL-1772) were maintained in DMEM media (Thermo Fisher Scientific) containing 4.5 g/L glucose, 110 mg/L sodium pyruvate, 10% FBS, 4 mM *L*-glutamine, and 1% streptomycin-penicillin. ARPE-19 retinal pigmented epithelial cells (male, RRID: CVCL_0145, Zoghbi lab, BCM, ATCC CRL-2302) were maintained in DMEM/F12 medium (Thermo Fisher Scientific) containing 3.2 g/L glucose, 10% FBS, 2.5 mM *L*-glutamine, and 1% streptomycin-penicillin. Tet21/N cells (female, RRID: CVCL_9812, Perini lab, University of Bologna, Italy) were maintained in DMEM media (Fisher Scientific) containing 4.5 g/L glucose, 10% FBS, 4 mM *L*-glutamine, and 1% streptomycin-penicillin under geneticin selection (0.2 mg/ml; G418, neomycin, Santa Cruz, CA, USA) and hygromycin (0.15 mg/ml; Sigma, MO, USA). All cell lines were validated by short tandem repeat analysis within the past 12 months and regularly tested for mycoplasma.

To knock down MYCN in LAN5 shMYCN cells, DOX (Santa Cruz) was added at a final concentration of 1 µg/mL for 72 or 96 h. To induce MYCN in MYCN3 cells, DOX was added at a final concentration of 1 µg/mL for 48 or 72 h. To turn off MYCN in Tet21/N cells, DOX was added at a final concentration of 2 µg/mL for 24 h. To activate MYCN-mediated transcription, SK-N-AS MYCN-ER™ cells were treated with 5 nM–1 µM 4-OHT (Sigma) for 1–6 days. Expression of MYCN and MYCN targets was verified before using the cell lines for further experiments.

To determine the role of exogenous FAs on cell growth, LAN5, IMR32, SK-N-BE(2c), SHEP, SK-N-AS and SK-N-AS MYCN-ER™ cells were cultured in complete media containing 10% FBS (Capricorn Scientific, Germany, FBS-12A) or in delipidized media containing 10% delipidized FBS (Capricorn Scientific, FBS-DL-12A) for up to 6 days. An FA mixture (Sigma, F7050) was added to delipidized media to rescue the effects of lipid deprivation. Drugs: A939572 (APExBIO, TX, USA, B3607), orlistat (Sigma, O4139), CB16.2 (ChemBridge, CA, USA, 5830995), CB5 (ChemBridge, 5674122), VP16 (Selleck Chemicals, TX, USA, S1225) and TMZ (Selleck Chemicals, S1237).

**Plasmids.** MYCN3 (MYCN Tet-On): To generate MYCN3 cells, *MYCN* cDNA was cloned into a pTR2-Hygro vector (BD Biosciences, NJ, USA) containing a tetracycline-responsive promoter. The construct was then transfected into a SHEP subclone stably expressing the tetracycline response element and selected with hygromycin. LAN5 shMYCN: GIPZ human MYCN lentiviral vectors (V2LHS_36755 and V3LHS_322662) were obtained from the Cell-Based Screening

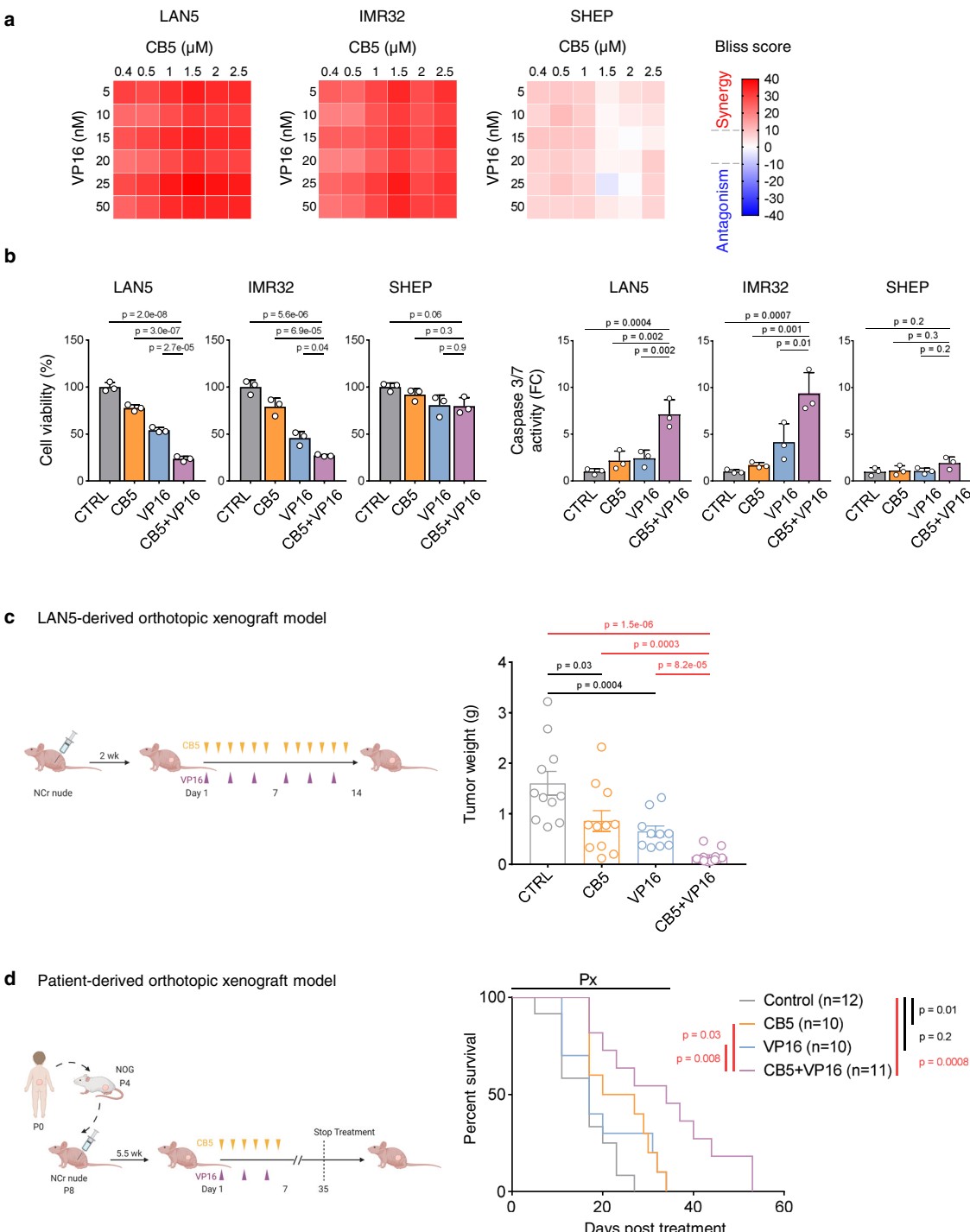

**Fig. 6 Suppressing FA uptake promotes the efficacy of conventional chemotherapies. a** Synergy analyses in MNA (LAN5 and IMR32) and non-MNA (SHEP) cells treated with CB5 (0.4–2.5 μM), VP16 (5–50 nM) and their combination for 72 h. Heatmap represents mean Bliss scores from three independent experiments. Bliss score >10 indicates synergy. **b** Cell viability and Caspase 3/7 activity after single and combination treatments. Cells were treated with CTRL, CB5 (3 μM), VP16 (80 nM), or their combination for 72 h. Mean ± SD ($n$ = 3); one-way ANOVA with Tukey's multiple comparisons test. **c** Anti-tumor activity of CB5 + VP16 combination therapy in LAN5-derived orthotopic xenografts. LAN5 cells were implanted in the renal capsule of NCr nude mice. Two weeks after implantation, mice were treated with CTRL (vehicle), CB5 (15 mg/kg, *b.i.d.* 6 days/week), VP16 (8 mg/kg daily, 3 days/week) or their combination for 2 weeks. Tumor weights after 2 weeks of treatment are shown. Mean ± SEM (CTRL = 11, CB5 = 11, VP16 = 10, CB5 + VP16 = 12); two-sided unpaired Mann–Whitney test. **d** Survival analysis in patient-derived orthotopic xenografts. Cells prepared from one MNA patient tumor (P8) were implanted in the renal capsule of NCr nude mice. Five and half weeks after implantation, mice were treated with CTRL (vehicle), CB5 (15 mg/kg, *b.i.d.* 6 days/week), VP16 (6 mg/kg daily, 3 days/week), or their combination for 5 weeks. Survival was plotted as a Kaplan–Meier curve and analyzed by the log-rank test (CTRL = 12, CB5 = 10, VP16 = 10, CB5 + VP16 = 11). Px treatment period; FC fold change. Source data are provided in the Source Data file.

Service (C-BASS) core at BCM. V3LHS_322662 (with better KD efficiency) was inserted into pINDUCER11 (C-BASS), containing a constitutive cassette (rtTA3 and eGFP) to detect infection efficiency and a turboRFP (tRFP)-shRNA cassette activated upon DOX treatment. sh*SLC27A2*: To knock down *SLC27A2* expression, two GIPZ lentiviral shRNA vectors with GFP reporters (V2LHS_27492 and V3LHS_398610, GE Healthcare Dharmacon, CO, USA) were used in LAN5 cells. V3LHS_398610 (with better KD efficiency) was used in IMR32 and SK-N-AS cells. FATP1 or FATP2 overexpression: FATP1 (C-BASS) or FATP2 (OriGene) cDNAs were cloned into pINDUCER20, a Tet-inducible lentiviral vector for ORF expression. Second generation lentiviruses were prepared as previously described[64].

**Mouse models.** In vivo studies were approved by the BCM Institutional Animal Care and Use Committee (AN7089 and AN6190). Mice were housed at the TCH Animal Facility, a temperature $(21 \pm 1^\circ C)$ and humidity (60%)-controlled and specific pathogen-free environment under a 14 h:10 h light/dark cycle. Mice were fed standard chow diet (LabDiet, 3002906-704) *ad libitum*. The maximal tumor size permitted was 1.5 cm by diameter, and this size was not exceeded in this study. *TH-MYCN transgenic model*: TH-MYCN$^{+/-}$ mice (129×1/svj), kindly provided by the Weiss lab (University of California San Francisco, CA, USA), were bred and pups were genotyped to identify TH-MYCN$^{-/-}$, TH-MYCN$^{+/-}$, and TH-MYCN$^{+/+}$ mice. TH-MYCN$^{+/+}$ 129×1/svj mice develop tumors (almost 100% incidence) at week 3–4 and succumb to disease by week 7–8[3]. *NB cell-derived orthotopic xenograft model*[65]: An inoculum of $10^6$ NB cells was surgically injected under the renal capsule of 6-week-old female NCr nude mice (CrTac:NCr-*Foxn1$^{nu}$*, Taconic, NY, USA). *TH-MYCN$^{+/+}$-derived orthotopic allograft model*: An inoculum of $10^6$ cells prepared from one TH-MYCN$^{+/+}$ tumor were injected under the renal capsule of 6-week-old female NCr nude mice (CrTac:NCr-*Foxn1$^{nu}$*). *TH-MYCN$^{+/+}$-derived orthotopic syngeneic model*: An inoculum of $10^6$ TH-MYCN$^{+/+}$ tumor cells from a TH-MYCN$^{+/+}$ tumor (different tumor from allograft model) was injected under the renal capsule of 6-week-old female TH-MYCN$^{-/-}$ mice (129×1/svj). *Patient-derived orthotopic xenograft model*: a primary MNA NB tumor (male, stage 4, P0) was implanted into the renal capsule of 6–8-week-old female NOG mice (P1, NOD.Cg-*Prkdc$^{scid}$ Il2rg$^{tm1Sug}$*/JicTac, Taconic). The NB tumor was verified histologically, then passaged in 6 to 8-week-old female NOG mice (NOD.Cg-*Prkdc$^{scid}$ Il2rg$^{tm1Sug}$*/JicTac) to P4 (Vasudevan lab, BCM), and finally passaged in 6 to 8-week-old female NCr nude mice (CrTac:NCr-*Foxn1$^{nu}$*, $2 \times 10^6$ cells) to P6 − 8 before initiating drug studies. Tumor engraftment and growth were monitored by bioluminescent imaging (Xenogen IVIS 100 System, Caliper Life Sciences, MA, USA) or MRI (echo time = 80 ms, repetition time = 3030 ms, slice thickness = 1.2 mm, field of view = 80 mm, number of slices = 18, matrix = $256 \times 250$, number of signal averages = 2, dwell time = 25 μs, scan time = 3.5 min, 1.0 T permanent MRI scanner, M2 system, Aspect Technologies, Israel). Bioluminescent images were analyzed by Living Image 4.7.3, and data are expressed as total flux. MRI images were analyzed and processed in Osirix (5.8.5, Pixmeo, Bernex, Switzerland), and data are expressed as tumor volume in cm³. *Power Analysis*: based on our previous studies[23], at least six mice per group are required to reach 80% statistical power at $p < 0.05$ (https://www.stat.ubc.ca/~rollin/stats/ssize/n2). Mice were randomized and evenly allocated into treatment groups. Non-engrafted mice were excluded from data collection. *Vehicle composition*: 10% polyethylene glycol 400 (Sigma, 202398), 10% Tween 20 (Fisher Scientific, BP337-500), and 80% PBS (Lonza, 17-516 F). CB5 (15–30 mg/kg *b.i.d.*) was administered *i.p.* for 2–6 weeks. VP16 (6–8 mg/kg) was administered *i.p.* three times per week for 2–5 weeks. At the end point (tumor diameter > 1.5 cm), mice were euthanized to determine tumor weight or survival. Intratumoral lipids were assessed by Oil Red O staining (Pathology Core, TCH). Briefly, frozen tumor sections were fixed in 60% iso-propanol for 15 min before staining with 0.003% Oil Red O solution (Sigma, O0625) for 30 min. Nuclei were counterstained with hematoxylin (Richard-Allen Scientific, MI, USA, 7211) for 30 sec. The total intensity of LDs was quantified by ImageJ2 (https://imagej.net/ImageJ2) from two representative images per tumor (blinded pathology review) and normalized by cell number. Samples with poor staining quality or without tumor areas were excluded from the ImageJ2 analysis. To assess tumor proliferation, apoptosis and MYCN expression, paraffin-embedded tumor sections were blocked with horse serum (10%) and incubated with Ki67 antibody (1:50, Biocare Medical, CRM325A, clone SP6, RRID: AB_2721189; Agilent, M7240, clone MIB1, RRID: AB_2142367), cleaved Caspase-3 antibody (1:400, Cell Signaling, 9661 L, RRID: AB_2341188) and MYCN antibody (1:100, Sigma, OP13-100UG, clone NCM II 100, RRID: AB_213284) at 4 °C overnight. Sections were washed with PBS and incubated with biotinylated anti-rabbit IgG (1:200, Vector Laboratories, BA1000, RRID: AB_2313606) and anti-mouse IgG (1:200, Vector Laboratories, BA9200, RRID: AB_2336171) antibodies at room temperature for 30 min, followed by incubation with 3,3'-diaminobenzidine and counterstaining with hematoxylin (Fisher Scientific, 7211). Five representative fields were captured by Nikon ECLIPSE 90i (x40, blinded pathology review). Ki67 positive cells, cleaved Caspase-3 positive cells, MYCN expression intensity, and the total number of cells were analyzed by ImageJ2. Data are presented as the percentage of Ki67 or cleaved Caspase-3 positive cells and MYCN intensity per cell. Samples with poor staining quality or without tumor regions were excluded from the ImageJ2 analysis.

## In vitro functional assays

*Cell viability*. Cell viability was determined using a Cell Counting Kit-8 (CCK, Dojindo Molecular Technologies, MD, USA, CK04-05) according to the manufacturer's instructions. For synergy studies, cells were treated with different doses of two drugs either alone or in combination. Cell viability was converted to inhibition % = 100%−viability% and synergy scores were calculated using the Bliss model (>10 indicates synergy; <−10 indicates antagonism)[53]. Synergy heatmaps were prepared in GraphPad Prism (7.01). *Caspase 3/7 activity*: Caspase 3/7-mediated apoptosis was measured using a Caspase-Glo assay kit (Promega, WI, USA, G8091) as previously described[23]. *FA uptake assay*: Real-time FA uptake was determined using a QBT Fatty Acid Uptake Assay Kit (Molecular Devices, CA, USA, R6132) according to the manufacturer's instructions. To determine compensatory FA uptake induced by FA synthesis inhibition, cells were serum-starved for 48 h, treated with 0–15 μM A939572 for 15 min, and a proprietary assay reagent was added. Plates were read at $\lambda_{ex}$ = 485 nm and $\lambda_{em}$ = 515 nm every 30 s for 30 min. To determine drug specificity, IMR32 FATP1 and FATP2-overexpressing cells were serum-starved and induced (DOX 20 ng/mL for 48 h or 72 h) before adding the assay reagent containing CB5 (0–6 μM). Plates were read at $\lambda_{ex}$ = 485 nm and $\lambda_{em}$ = 515 nm every 30 sec for 100 min. FA uptake imaging was performed by fluorescence staining of an FA analog (BODIPY™ 558/568 C12, D3835 and BODIPY™ 500/510 C1, C12, D3823, Thermo Fisher) using an Olympus IX71 (Olympus, Japan). The staining solution contained 10 μM fluorescence dye, 5 μM BSA and 0.2% trypan blue in phenol red-free delipidized RPMI media, modified from Li et al.[66]. Images were quantified by ImageJ2: corrected total cell fluorescence (CTCF) = integrated density−(area of selected cell×mean fluorescence of background readings)[67]. *Clonogenic assay*: LAN5 shCTRL and sh*SLC27A2* cells were seeded at low density on a 6-well plate. After 12 days of culture, with media replaced every 3 days, the plate was stained with 0.25% Coomassie Brilliant Blue G (Sigma, 27815) for 20 min at room temperature and rinsed with distilled water to remove excess dye. The plate was then imaged using a FluorChem R System (ProteinSimple, CA, USA) and the number of colonies was counted using alpha View-FluorChem Q software (3.4.0.0).

## Real-time qPCR and western blotting

*qPCR analysis*. Total RNA isolated using an RNeasy Mini Kit (Qiagen, MD, USA, 74104) was directly mixed with the reagents supplied in the QuantiTect SYBR® Green RT-PCR Kit (Qiagen, 204243) and subjected to one-step RT-PCR performed on StepOnePlus™ Real-Time PCR System (Thermo Fisher Scientific). Primer sequences are listed in Supplementary Data 4. *Western blotting*: Cells and tumors were lysed with RIPA buffer (Thermo Fisher, 89900) containing protease inhibitors (Roche, 04693116001). Protein concentrations were measured by Bradford assay and 50–75 μg protein was electrophoresed and transferred. Primary antibodies: MYCN (1:500, Cell Signaling, 9405 S, RRID: AB_10692664), FATP1 (1:500, Abcam, ab69458, RRID: AB 1270734), FATP2 (1:500, Abcam, ab83763, RRID: AB 1859828; Thermo Fisher Scientific, PA5-30420, RRID: AB 2547894; Thermo Fisher Scientific, PA5-42429, RRID: AB 2610399), total and cleaved PARP (1:500, BD Biosciences, 551024, clone 7D3-6, RRID: AB_394008), total Caspase-3 (1:500, Santa Cruz, sc-65497, clone 4.1.18, RRID: AB 1120001), total cleaved Caspase-3 (1:300, Cell Signaling, 9661 S, RRID: AB 2341188), p21 (Waf1/Cip1) (1:300, Cell Signaling, 2947 S, clone 12D1, RRID: AB 823586), p53 (1:500, Santa Cruz, sc-126, clone DO-1, RRID: AB 628082), SCD1 (1:1000, Cell Signaling, 2438 S, RRID: AB 823634), ACC (1:1000, Cell Signaling, 3662 S, RRID: AB 2219400), CypB (1:500, Santa Cruz, sc-130626, clone k2E2, RRID: AB 2169421), and ACTB (1:5000, Sigma, A2228, clone AC-74, RRID: AB 476697). Secondary antibodies (LI-COR Biosciences, NE, USA): IRDye® 680RD Goat anti-mouse IgG (1:10000, 925-68070, RRID: AB 2651128), IRDye® 800CW Goat anti-mouse IgG (1:10000, 926-32210, RRID: AB 621842), and IRDye® 800CW Goat anti-rabbit IgG (1: 10000, 926-32211, RRID: AB 621843). Membranes were scanned using an Odyssey Infrared Imaging System (Odyssey® Application Software 3.0, LI-COR Biosciences).

## Metabolomics, lipidomics, and FA analyses

*Global metabolomics*. Cell samples (MYCN KD 0, 72, 96 h; MYCN-ON 0, 48, 72 h; $n = 4$ per condition) and primary tumor samples (MNA, $n = 18$; non-MNA, $n = 18$) were submitted to Metabolon Inc. for untargeted metabolomic profiling (DiscoveryHD4 Metabolic Platform, $n = 545$ compound library, NC, USA). One-way ANOVA was used to compare metabolite levels between groups. Welch's two-sample $t$-test was used to compare metabolite levels between MNA and non-MNA primary tumors ($p$-value ≤ 0.05). The metabolite classification network is presented by Cytoscape 3.8.2 using subpathways as the source nodes and metabolites as the target nodes. Target node sizes indicate the absolute log$_2$FC of metabolite levels between MYCN KD 72 h vs. CTRL, MYCN-ON 72 h vs. MYCN-OFF and MNA vs. non-MNA. Red: upregulated metabolites; blue: downregulated metabolites ($p$ ≤ 0.05); gray: no significant change ($p$ > 0.05). To determine metabolic changes in each subpathway, a differential abundance score was calculated for each condition (adapted from Hakimi et al.[45]). Differential abundance score = (# of significantly increased metabolites− # of significantly decreased metabolites) / # of metabolites measured in subpathway × 100%. Differential abundance scores were calculated in subpathways containing at least three differentially altered metabolites in at least one condition ($p$ ≤ 0.05). A score of 100 indicates that all

metabolites in a subpathway are upregulated; a score of −100 indicates that all metabolites in a subpathway are downregulated ($p \leq 0.05$). To identify subpathway enrichment, two enrichment algorithms were used. Algorithm 1: Over-representation analysis using the hypergeometric distribution with $p$-values adjusted by the Benjamini–Hochberg procedure. "*" Indicates that a subpathway is significantly enriched in one comparison group (adjusted $p$-value < 0.25), and subpathways were ranked by the number of "*" designations. Subpathways with the same number of "*" designations were ranked by the average absolute differential abundance score from three comparison groups. Algorithm 2: GSEA (4.0.3)[68] used subpathways as the 'gene' set reference and included subpathways containing at least three metabolites. GSEA was conducted with 10,000 randomized metabolite sets to estimate statistical significance, and the signal-to-noise metric (Z-score) between the two phenotypes was used to rank all metabolites. Subpathways with FDR < 0.25 were selected for presentation.

*Lipidomics*. Briefly, lipid extracts ($n \geq 4$ per group) were analyzed by liquid chromatography (LC)/triple time-of-flight (TOF) (Shimadzu CTO-20A Nexera X2 41 UHPLC system, Kyoto, Japan)[23]. Mass spectrometry (MS) analysis was conducted on a Triple TOF 5600 equipped with a Turbo V™ ion source (AB Sciex, Concord, Canada). Data were acquired with Analyst TF software 1.8 (AB Sciex). Data were normalized by median interquartile range normalization and were $\log_2$ transformed. Student's $t$-test was used to compare differences between two groups. $P$-values were adjusted by the Benjamini–Hochberg procedure to obtain FDR. Changes with FDR < 0.25 (LAN5 shMYCN and MYCN3 cells) and with absolute $\log_2$FC > 1 and FDR < 0.25 (SK-N-AS MYCN-ER™ cells; LAN5 shCTRL and sh*SLC27A2* tumors) are presented in heat maps (R software 4.0.4).

*FA profiling*. Intracellular total FAs were analyzed by either LC-MS ($n = 4$ per group) or gas chromatography (GC)-MS ($n = 6$ per group). For LC-MS analysis, extracted FAs were analyzed by high-performance LC coupled with Agilent 6495 QQQ MS (Agilent Technologies, CA, USA) using ESI negative ionization via single-reaction monitoring. Data analyzed with Agilent mass hunter quantitative software (10.0) were normalized with an internal standard and $\log_2$ transformed per sample. For GC-MS, FA concentration and tracer enrichment were measured using pentaflurobenzylbromide derivatization and GC-MS negative chemical ionization as previously reported[69], with a slight modification: an SP-2380 column (Supelco, PA, USA) was used to improve the separation of palmitoleic acid (16:1n7) and sapienic acid (16:1n10) peaks. Data were acquired in selective ion-monitoring mode. Analyte and standard peak areas were determined. The ratio of the area from all (M0-M2) analyte-derived ions relative to that of the internal standard tridecanoic acid (m/z, 213–215) was calculated. Ratios were compared with the calibration curves of serial FA concentrations to determine each FA concentration. Student's $t$-test was used to compare FA levels between groups. $P$-values were adjusted using the Benjamini–Hochberg procedure to obtain FDR. Changes with FDR < 0.25 were selected for data presentation.

*FA tracing*. Briefly, deuterated water ($D_2O$) (99 atom%, Cambridge Isotope Laboratory, MA, USA) was used to determine de novo lipogenesis[23]. [U-$^{13}$C] palmitic acid (16:0) potassium salt (99 atom%$^{13}$C, Cambridge Isotope Laboratories) was used to determine FA desaturase activity (ratio of [$^{13}C_{16}$]monounsaturated FA to [$^{13}C_{16}$]saturated FA) and FA uptake (original [$^{13}C_{16}$]FAs in medium—remaining [$^{13}C_{16}$]FAs in medium[$^{13}C_{16}$]). FAs were analyzed by GC-MS and normalized by the total protein contents of each sample.

### Chromatin immunoprecipitation (ChIP) and luciferase activity assay

*MYC(N) ChIP*. Briefly, 1x10$^7$ cells were cross-linked with 1% formaldehyde, and the reaction was stopped with 0.125 M glycine before harvest. DNA was sheared by sonicating using Bioruptor PLUS (Diagenode, NJ, USA). A small aliquot of sonicated material was retained as input and the remaining sample was immunoprecipitated using 5 μg ChIP-grade antibodies (MYCN, Santa Cruz, sc-53993, clone B8.4.B, RRID: AB 831602; c-MYC, Santa Cruz, sc-764, clone N-262, RRID: AB 631276). Rec-sepharose protein A beads (Invitrogen, CA, USA, 101141) were used to immobilize immuno-complexes. Crosslinking was reversed using RNase-A treatment (37 °C, 1 h) and proteinase K (Roche) for 6 h at 65 °C. Immunoprecipitated DNA was purified by phenol/chloroform and ethanol precipitation. MYCN and c-Myc binding to the *SLC27A* promoter was analyzed by qPCR (SSOAD-VANCE-BIORAD) using primers listed in Supplementary Data 4 (amplicon regions included), and normalized by the fold enrichment (2$^{-\Delta\Delta CT}$) method using *ABCA10* TSS as a negative control. *Dual luciferase reporter assay:* The effects of MYCN on *SLC27A1* and *A2* promoter activities were analyzed using a dual luciferase gene reporter assay as previously described[70]. Luciferase reporter activity was measured using the Dual Luciferase Assay System (Promega, E1980). Chemiluminescence values for firefly and renilla luciferases were measured using a GloMax 20/20 instrument (Promega), and data are reported as percentage of the MYCN-OFF/MYCN-ON ratio in Tet21/N cells. MYCN expression was turned off (DOX 2 μg/mL for 24 h) 6 h after transfection (Effectene, Qiagen), and activities were compared with those in untreated cells. Empty vector (pGL3b) and pGL3-*ODC1* promoter were used as negative and positive controls, respectively. pGL3 constructs

carrying *SLC27A1, SLC27A2* and *ODC1* promoters were obtained by directional cloning using primers listed in Supplementary Data 4 (promoter regions included). An E-BOX *SLC27A2* mutant (from CACCTG to GAATTC) promoter construct was obtained by whole-around PCR technology using primers listed in Supplementary Data 4.

**Data and statistical analysis**. *SLC27A2* dependency scores were extracted from the CRISPR (Avana) Public 20Q4V2 dataset[49], which is part of the DepMap project (The Broad Institute, USA). This dataset was generated from a CRISPR knockout screen of 18,119 genes in 789 cell lines across 30 primary diseases. A low dependency score indicates a high dependency on *SLC27A2* for survival.

Microsoft Excel (2013) was primarily used for data collection, and GraphPad Prism (7.01) and R (4.0.4) were used for data analyses, unless otherwise specified. BioRender.com was used to generate diagrams in Figs. 1a, 5a, d, h, i, 6c and d. Investigators were blinded to cell identity during metabolomics, lipidomics, FA profiling, and pathological analyses. For in vitro experiments, we assumed a normal distribution and performed two-sided unpaired parametric tests (Student's $t$-test) for two-group comparisons. Multiple-group comparisons used one-way or two-way ANOVA with Dunnett's or Tukey's multiple comparisons test. For animal studies, differences in tumor incidence between groups were computed by Fisher's exact test. Tumor growth and mouse weights were analyzed by two-way ANOVA followed by Sidak's multiple comparisons test. Tumor weights and protein markers were compared using two-sided unpaired Mann–Whitney tests. Survival was plotted using Kaplan–Meier curves and analyzed using the log-rank test. Data are shown as the mean, mean ± SD or mean ± SEM ($n$ indicates number of biologically independent samples and is provided in the figure legend). $P$-values < 0.05 were considered significant.

**Reporting summary**. Further information on research design is available in the Nature Research Reporting Summary linked to this article.

## Data availability

Five human NB tumor datasets (Kocak, GSE45547, $n = 649$; NRC, GSE85047, $n = 283$; Lastowska, GSE13136, $n = 30$; Hiyama, GSE16237, $n = 51$; and Versteeg, GSE16476, $n = 88$) and three normal tissue datasets (Adrenal Gland, SN_ADGL, $n = 13$; Neural Crest, GSE14340, $n = 5$; and Normal Various, GSE7307, $n = 504$ including 108 types of normal tissues) were included in this study. They are all publicly available at NCBI Gene Expression Omnibus or R2: Genomics Analysis and Visualization Platform. CRISPR (Avana) Public 20Q4V2 and the long-chain FA transport geneset (GO: 0015909) are also publicly available. Raw data and uncropped blots associated with Figs. 1–6 and Supplementary Figs. 1–7 are provided as a single Source Data file. Patient sample information, metabolomics and lipidomics data, as well as oligonucleotide information are available in the Supplementary Files. Source data are provided with this paper.

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

## Acknowledgements

We acknowledge Philip Bocock and Scott McCulloch from Metabolon, Inc. for their support with sample preparation and data analysis. We thank Alessandro Baldan for providing primary samples (RTSS, TCH), Brian J. Altman (University of Rochester) for providing SK-N-AS MYCN-ER™ cells, Ronald Bernardi (BCM, now at Genentech) for providing SK-N-BE(2c) and LAN5 shMYCN cells, Michele Redell (BCM) for providing HS-5 cells, Joel R Neilson (BCM) for providing C2C12 cells, and Huda Zoghbi (BCM) for providing ARPE-19 cells. We thank our master student Nathan Drolet (BCM), summer student Kevin Rodriguez (University of Houston) and visiting post-doc Davide Leardini (University of Bologna) for their technical support. We also thank the Cell-Based Assay Screening Service (BCM), the Stable Isotope Core Laboratory (CNRC, BCM), the Metabolomics Core (BCM) and the Pathology Core (TCH) for their technical support. L.T. is supported by Kate Amato Foundation. E.B. is supported by NIH R01 (R01-CA222224), DOD (W81XWH-19-1-0556), and CPRIT (RP180851). P.S. is partially funded by European Commission, HORIZON 2020 (826121). Y.Z. is partially supported by NIH grant R01CA224304. K.B.G. and A.B. are supported by NIH (NCI/NIDCR 1U01DE028233), St. Baldrick's Research Grant (714511) and the Russell and Glenda Gordy Center for Innovative Therapies at Texas Children's Cancer Center. N.P. is supported by NIH/NCI R01CA220297 and NIH/NCI R01CA216426. G.P. is supported by Italian Association for Research on Cancer (AIRC2020-IG24341). T.D.P. and C.C. are partially supported by the NCI P30 shared resource grant (CA125123), CPRIT (RP170005 and RP200504), and NIEHS P30 (ES030285) and P42 (ES0327725) grants. This project is supported by CPRIT Proteomics and Metabolomics Core Facility (RP210227), NIH (P30 CA125123) and Dan L. Duncan Cancer Center.

## Author contributions

L.T. and E.B. formulated the ideas and aims of this study. L.T., M.A.M., G.M., M.M.-S., A.B., B.E.H., A.B.W., and S.A. performed the experiments. L.T., M.A.M., G.M., M.M.-S., T.D.P., B.Z., A.B.W., Z.Z., J.H.F., J.H., C.C., and E.B. analyzed the data. P.S., Y.Z., K.B.G., N.P., G.P., C.C., and E.B. supervised the study. L.T. and E.B. wrote the manuscript with input from all authors.

## Competing interests

The authors declare no competing interests.
