## [Peer Review File · Nature Communications]

Title: MYCN-driven Fatty Acid Uptake is a Novel Metabolic Vulnerability in NeuroblastomaREVIEWER COMMENTS

Reviewer #1 (Remarks to the Author):

The paper by Ling Tao et al. is a comprehensive metabolomic study involving both cell and mouse models with overexpressed or knocked-down MYCN, as well as patient samples, to identify a mode of action of MYCN in neuroblastoma. Myc has previously been linked to lipid and fatty acid metabolism in other oncologies but here a highly specific FA transporter is identified and shown to be a pharmacological target, with an inhibitor demonstrating efficacy in orthotopic patient-derived tumour models.

It is generally well written and the data clearly presented.

There are minor modifications required prior to publication.

Line 127

“in two MYCN-induced NB cell line models and in NB primary tumors”

One model is Tet-ON Dox-induced, while the other here is conditionally knocked down by Dox-inducible shRNA expression. Perhaps a better way to describe them therefore is “conditionally-expressed MYCN”, or other way to reflect that they are not MYCN-induced lines per se.

Line 194

“These findings were validated in a second MYCN system, SK-N-AS MycN-ER.”

Since this is a new assay (lipid droplets) being introduced to extend the FA phenotype to an additional cell type, why isn't this assay performed in parallel with MYCN3-Tet-ON cells to show that they respond similarly?

Fig 2A

This appearance of lipid droplets after MYCN activation contrasts with the report from Marie Arsenian Henriksson where such droplets were observed after Myc inhibition in neuroblastoma (Zirath et al. 2013). Please briefly discuss this discrepancy.

Fig 2D

While the effects of FA supplementation on the MNA cells vs. SHEP are clear, for the SK-N cells with or without activated MycN-ER it is more subtle (although significant). It is a little confusing to represent the same data in a different format here (Fig 2D right panel vs. the other panels), and perhaps drawing the data in the same way may show how the MycN-ER cells are more similar to the MNA cell lines, whereas without Myc-ER they are more similar to SHEP cells (therefore, plot the right panel of Fig 2D in 2 separate line graphs, not as single bar graph).

Line 286

“supporting that NB cells highly depends”

The grammar requires correction here.

Fig 4J

There is no legend for the blue and black bars (in fact, they are coloured incorrectly and should be red, presumably).

Line 425-6

“These effects can be rescued by FA supplementation.”

From Fig 2D, it seems that there is only a partial rescue, and in fact in IMR32 and MYCN-ER cells, it is a relatively small effect. This should be reflected in the text here (since as written it implies a complete rescue) plus one could speculate as to why there is not a greater rescue with FA supplementation. This extends to Line 230 too (“Collectively, our data indicate that MNA-induced cell growth critically depends on exogenous FAs.”). Again, if it is critically dependent, why isn’t the rescue from FA supplementation more notable? It is a modest – although significant – effect in Fig 2E (right panel) for MYCN-ER active cells vs. inactive.

Reviewer #2 (Remarks to the Author):

This is a very interesting and ambitious study proposing a new approach for neuroblastoma therapy based on the identification of a novel metabolic target . However, there are several concerns that should be addressed before acceptance.

Major concerns:

1. Please show WB on MYCN, proliferation, and apoptosis markers after treatment with the inhibitors in the cell lines. In addition, authors only show changes in MYCN expression in LAN5 and MYCN3 in SFigure 1A, but not in any other of their cell lines after induction/ repression of MYCN.
2. The authors have analyzed lipid accumulation in some of their experiments. It has previously been shown that inhibition of MYCN results in lipid accumulation see for instance Zirath PNAS, 2013. How does the lipid accumulation occurring after CB5 treatment correspond to the lipid accumulation caused by MYCN inhibition?
3. Figure 4 and legend. Photos of LAN5 and SHEP cells are shown after BOIDPY. It is obvious by eye that the fluorescence is decreasing more in the LAN5 cells but is also clearly decreasing in the SHEP cells. However, in the graph it looks more or less as if there is no decrease in the SHEP cells. In addition, it is indicated that * means significant changes after CB5 treatment and that # indicates significant differences between LAN5 and SHEP cells at same CB5 dose. Please clarify. The use of # is unusual.
4. Lines 407-410 “Our metabolomics data show reduced glutamine and glutamate in MYCN-overexpressing cells and MNA tumors, suggesting enhanced consumption of glutamine and glutamate. Future research should include in vivo imaging and flux studies to better understand MYCN reprogramming of glutamine metabolism.” Other studies have demonstrated that MYCN de novo

glutamine synthesis. Please mention and give reference.

5. The authors confirm that SLC27A2 is required for NB survival performing shRNA in LAN5 cells. These results should be validated in a second MNA NB cell line. Moreover, to prove that it is a target of MYCN and not c-MYC, authors should also prove that shSLC27A2 does not affect viability in non-MYCN NB cells.

6. Author demonstrated lipid droplet accumulation as a result of increased fatty acid uptake and synthesis was demonstrated in SK-N-AS MYCN-ER cells. Authors should validate that this is a result due to MYCN, by repeating this experiment in another cell line, for instance the MYCN3, in which they performed the tracing experiments. This will prove that lipid accumulation is not due to treatment with 4-OHT which could induce stress.

7. In the Figure legend for Figure 4 E-G, it is mentioned ctrl versus CB5 while the Figure 4 E-G shows ctrl versus shSLC27A2. Please revise.

8. Line 244. Specify the coordinates of the region analyzed for SLC27A2 in the ChIP-qPCR experiments. Are there any E-boxes in that region? It would also be useful to check available ChIP-seq data in the ENCODE database and see if c-MYC also binds to the promoter of SLC27A2.

9. The authors show a non-canonical E-box in the promoter of SLC27A2 and show MYCN binding to the promoter. To show that it is a target gene authors need to mutate the E-Box and analyze binding by ChIP. Furthermore, both wt and mutant E-box-reporters, for instance luciferase, and assays should be performed to validate transcriptional activation of SLC27A2 by MYCN.

10. The major concern is the lack of biological replicates in many of the in vitro experiments shown in this manuscript. In the materials and methods section, the authors mention that at least six mice are necessary for in vivo experiments to reach an 80% statistical power at $p < 0.05$. But for several of the in vitro experiments, the authors only present two replicates. Statistical analysis cannot be performed with a $n < 3$. Authors should discuss how the statistical analysis was performed and represent experiments with at least three times for significance. It is important to present at least three replicates per experiment with the corresponding statistical analysis. This concerns the following Figures: Fig 2B ($n = 2-3$), 2C ($n = 2$), 2D ($n = 2-4$), 2E ($n = 2$), 3C ($n = 2$), 4D ($n = 2$), and 6B ($n = 2$). Supplementary Figures 2B ($n = 2-4$), 2C ($n = 2$), 2D ($n = 2$), 3B ($n = 2$), 4B ($n = 2-3$), 6B ($n = 2$).

11. The authors present data from seven orthotopic mouse models:

I. LAN5 shCTRL and shSLC27A2 cells were implanted in the renal capsule of NCr nude mice. MRI. Tumor weight. ORO-staining.

II. LAN 5-luciferase cells in NCr nude mice. Treated with vehicle or CB5. IVIS. Tumor weight.

III. TH-MYCN +/+ derived allograft model in NCr nude mice. Treated with vehicle or CB5. MRI. Tumor weight. ORO-staining.

IV. TH-MYCN+/+ derived syngeneic model in 129x1/svj mice. Treated with vehicle or CB5. Tumor weight.

V. MNA patient tumor. Treated with vehicle or CB5 after two weeks.

Six weeks. Survival analysis.

VI. LAN-5 cells in NRc nude mice. Treated with vehicle, CB5, VP16, or CB5/VP16.

Tumor weight. Survival analysis.

VII. MNA patient tumor. Treated with vehicle, CB5, VP16, or CB5/VP16 after 5,5 weeks. Seven weeks. Survival analysis.

Concerns re animal experiments:

11A. Please show the actual size of the tumors from the different mouse models so it is possible to compare them.

11B. Figure 5H. Authors have used mice with normal immune system and write that it is important that tumor growth can be reduced by inhibiting SLC27A2. Is it not the growth of the TH-MYCN spheres which is the major achievement in this experiment, ie. that tumors can form even in the setting of a normal immune system rather than that the tumor burden can be reduced? Comparing the graphs in Figure 5F with Figure 5H, respectively, show that the tumors in fact get larger in the syngenic model compared to in the nude mice during the same time frame. Do the authors have any idea why the tumors from the TH-MYCN+/+ cells become larger in the 129x1/svj compared to the NRC nude mice? Please comment on this in the discussion.

11C. The authors have analyzed some of the tumors generated for lipid accumulation but not stained for any other markers (Figure 4G and Figure 5G). Please show stainings of the tumors at endpoint for at least MYCN, proliferation, apoptosis and maybe also for hypoxia, and angiogenesis markers to understand more about the mechanism for reduced tumor growth. In addition, how did the authors choose the mice to analyze for lipid accumulation? Please explain why you chose to stain for lipids only in the LAN5 shCTRL and shSLC27A2 and in the TH-MYCN +/+ derived allograft model in NCR nude mice.

11D. Why did authors wait 2 weeks in the first and then 5,5 weeks in the experiments with the MNA patient tumors?

11E. The representation of tumor volume is unclear in Figure 5E. The authors show a significant difference between control and CB5-treated mice, however the graph show tumor volumes similar for both groups. Please, provide the average of each group.

11F. SFigure 5C and SFigure 5D. Why do the mice increase weight during the experiments with treatment with CB5 in TH-MYCN+/+ derived syngenic and in patient derived MNA xenograft models, respectively? Furthermore, no statistics is given in these experiments. The same is true for the experiments shown in SFigure 5A and SFigure 5B.

11G. Authors show that CB5 has minimal effect in non-MNA cells in vitro but they have not proved this in vivo. Treatment of mice implanted with non-MNA NB cells should be performed to further validate the in vitro results. Moreover, NCR nude mice were treated with 25 mg/kg b.i.d, while in the immunocompetent 129x1/svj mice the concentration of CB5 was 30 mg/kg b.i.d. Please, explain why the concentration is higher in the latter.

11H. The In vivo experiments show that the mice do not lose weight as a sign of toxicity derived from inhibition of SLC27A2. In the discussion it is mentioned that normal tissue toxicity is missing while inhibiting FA synthesis. Does inhibition of SLC27A2 affecting organs as the liver or kidneys?

11I. There are important differences between the mice treated with CB5 in Figure 5E but still the differences with the control group are significant at a p-value of 0.006. Please revise and provide a graph with the average and SD of all the mice. The present Figure 5E can be shown in SI.

Minor comments:

1. To avoid any confusions, authors should mention that SLC27A encodes FATP2 earlier in the text, also in the abstract. Along the same line, authors use abbreviations in the results part that are not spelled out. For easier reading of the manuscript it is preferable to spell out the abbreviations in the text rather than in the Figure legends. Some examples: Line 128 MNA and non-MNA, Line 150 KD, OE, Line 185 D2O, 4-OHT. In the animal experiments: b.i.d. Similarly, for the names of the metabolic enzymes: FASN, ACACA, and SCD1.

2. The following sentence should be revised, and relevant references cited, as there are many studies evaluating the metabolic changes induced by MYCN in different NB models and, also, in patient samples: “However, no studies have comprehensively evaluated the metabolic changes induced by MYCN across multiple NB models, including patient samples.”

Some recent references:

o Wang T, Liu L, Chen X et al. MYCN drives glutaminolysis in neuroblastoma and confers sensitivity to an ROS augmenting agent. *Cell Death Dis* 9, 220 (2018).

o Oliynyk G, Ruiz Perez M, Sainero-Alcolado L et al. MYCN-enhanced oxidative and glycolytic metabolism reveals vulnerabilities for targeting neuroblastoma. *iScience* 21, 188–204, (2019).

o Ding Y, Yang J, Ma Y, et al. MYCN and PRC1 cooperatively repress docosahexaenoic acid synthesis in neuroblastoma via ELOVL2. *Exp Clin Cancer Res.*38(1):498 (2019).

o Ruiz-Pérez MV, Sainero-Alcolado L, Oliynyk G, et al. Inhibition of fatty acid synthesis induces differentiation and reduces tumor burden in childhood neuroblastoma. *iScience* 24(2):102128 (2021).

3. The authors do not specify the time point in which the IC50 values in Figure 2C were calculated.

4. In line 246, Figure 3B is mentioned while it should be Figure 3C.

5. In Lines 352 and 353, Figure 5J is mentioned but there is no Figure 5J in Figure 5. Please revise.

6. The authors write “Our study provides the first direct evidence that MYCN-driven tumors rely on FA uptake and lipid synthesis for survival, making targeting of FA uptake a promising therapeutic approach for high-risk MNA patients.” Again, there are several articles also demonstrating this statement, so the words “first evidence” should be avoided. Authors use similar expressions elsewhere in the manuscript, please do not use “for the first time”, “first report” and similar expressions.

7. Lines 178-180. How do the authors explain the mechanism of MYCN promoting FA desaturation? Does MYCN regulate any enzyme involved in that process?

8. Figure 6A-B and Supplementary Figure 6A-B. Please show the cell viability graphs for each cell line and treatment. SHEP cells do not show Caspase 3/7 activity, but we cannot know whether these compounds still affect cell viability. There is the possibility that the selected compounds induce cell death in SHEP by a mechanism other than apoptosis.

9. Figure 4J. The different concentrations are presented as white (0), grey (5 uM) and red (10 uM) CB5, but there are bars in the Figure with blue and black colors. Nothing is neither mentioned about the blue and black bars in the legend. Please advice and revise.

10. The authors are using several different inducible neuroblastoma cells for different assays. Please clarify the reasoning behind using these specific cells for the specific assays and if they have tested also some of the other cell lines in the experiments presented than the ones shown, with similar or with different results.

11. Please use italics when referring to the gene and for *in vitro*, *in vivo*, *i.p.*, *b.i.d.*, etc

12. Line 381 "All treatments did not cause significant body weight loss or clinical signs of toxicity." Revise to -> No treatment caused any significant body weight loss or clinical signs of toxicity.

13. Figure Figure 3C, SFigure 3B and SFigure 3C. Please specify the regions and the primers used for the PCRs.

Reviewer #3 (Remarks to the Author):

The manuscript by Tao and colleagues describes a multi-omics analysis of MYCN-amplified and non-amplified neuroblastoma (NB) systems, including patient-derived tumour samples. This analysis points towards NMY-dependent changes in glycerolipids, mostly triglycerides (TG) and diacylglycerides (DG). The authors then go on to show that MYCN-amplified NB cells are dependent on lipid uptake as they are more effected by lipid depleted serum or inhibition of fatty acid uptake than non-amplified counterparts. This is then linked to enhanced expression of the fatty acid transporter FATP2 (SLC27A2), which the authors identify as an MYCN-target gene by chromatin-IP and promoter studies. They also analyse public datasets derived from NB patients to show that SLC27A2 expression correlates with MYCN expression/activity and poor patient survival. They then use RNAi-mediated gene silencing to show that FATP2 is required for *in vitro* and *in vivo* growth of MYCN-amplified NB cells. They also perform lipidomics analysis of tumour material after SLC27A2 silencing to confirm that TG levels are reduced. In addition, they block the activity of FATP2 using the small molecule compound CB5 (also termed *grassofermata* in previous studies) to show reduced tumour growth and extended survival in several MYCN-dependent *in vivo* models. Finally, they show that CB5 synergises with two chemotherapeutic agents (VP16 and temozolomide) both *in vitro* and *in vivo*. They conclude that enhanced fatty acid uptake through induction of FATP2 promotes NB growth and therapy resistance.

Overall, the manuscript shows a plethora of data generated across multiple *in vitro* and *in vivo* systems. It makes a strong case for enhance fatty acid uptake in MYCN-amplified NB cells and tumours and provides evidence for treatment resistance, albeit without deeper mechanistic insight. There are a number of points that should be addressed before this manuscript can be considered for publication.

1) Several of the bar graphs shown in the Figures do not show error bars in the controls (i.e. Fig. 2E, 3B etc). Variability of the controls should also be displayed.

2) The different sensitivity of SK-N-AS NMYC-ER cells to medium lipid depletion is quite small. The cells without MYC activation are already quite sensitive to lipid depletion (similar to LAN5 cells). Is this explained by a leakiness in the induction system? How would parental SK-N-AS cells respond? Given that these cells are non-MYC amplified, a high sensitivity towards lipid depletion would be contradictory to the authors' conclusions.

3) The specificity of the inhibitors used in this study is purely based on literature evidence. As a large proportion of the in vivo findings rely on the CB5 compound, it would be good to establish at least some level of specificity of this drug in the systems used here. For example, the authors could monitor restoration of fatty acid uptake after overexpression of another fatty acid transporter that should not be affected by the compound.

4) The lipidome analysis of tumour tissue is somewhat confusing. The majority of lipids displayed in the heatmap in Fig. 4H, including phosphatidylcholine species, actually show a strong upregulation in SLC27A2-silenced tumours. Only DG and TG species seem to be selectively downregulated. Does this mean that fatty acids taken up by the tumour cells are funnelled specifically into the TG synthesis pathway? How does this compare to the lipid species identified to be altered in the initial metabolomics analysis performed on cell lines and tumours (Fig. 1)? This experiment needs deeper exploration and discussion.

5) In the orthotopic allograft model (TH-MYCN+/+ in NCr nude mice, Fig. 5D-G), there are two clear outliers in the CB5 treatment group. Where these tumours also analysed by Oil-Red-O staining? This would be important to judge whether failure to inhibit tumour growth could be due to poor availability of the drug in the target tissue or due to potential resistance mechanisms (i.e. activation of fatty acid synthesis). Would it be possible to confirm drug availability in the tumour tissue?

6) The manuscript is quite difficult to read. This is most likely due to that large amount of data but some editing for clarity would help making the study more accessible. In particular, the first paragraph of the discussion section should be reconsidered. The authors mention several isolated findings and speculate about potential mechanisms that were not investigated in the manuscript. This distracts from the discussion of the main conclusions from the study. In particular, the discussion of potential roles of polyamine and glutamine metabolism in MYCN-amplified NB does not add further insight. Moreover, the discussion refers to the downregulation of glycerophosphoglycerols by MYCN but this is not mentioned in the results section and cannot easily be delineated from the figures. The section on metabolism and immune function is also quite unclear as the authors report reduced tumour growth in both immunocompetent and deficient in vivo systems.

Specific points:

Line 101: Something seems to be missing from this sentence. Should this be in a new paragraph?

Line 128: The description of the experimental systems used in this study is a bit confusing. It should be made clear that MYCN overexpression (MYCN3 model) was performed in non-MYCN amplified SHEP cells.

Figure 1C: It would be useful to assign the indicators of significance to the respective comparison (i.e. tumours, overexpression or KD). Asterisks could be organised in a grid. It would be particularly interesting to see pathways that are only significantly regulated in tumours. Remarkably, only one pathway (mapping to DG metabolism) is significantly altered in all three experimental systems. This should be discussed.

Figure 1E: Why were these TG species (TG 42:0 and TG 48:2) chosen for display as box plots?

Line 154: GSEA on the same data cannot be used to “validate” findings. This should be rephrased.

Line 205: The statement that both A939572 nor CB5 have no cytotoxic effect but that CB5 is less toxic does not make much sense. This should be rephrased.

Line 246: Reference to figure should be 3C.

Line 317: The MNA LAN5 orthotopic implantation model has already been introduced on the previous page.

Line 453: The meaning of this sentence is unclear. Do the authors mean immune cell activity targeted against MYCN positive NB tumours or MYCN function in immune cells.

Line 455: Insert new paragraph after “therapeutic targets”?

Line 461: Typo in temozolomide.

Dear Reviewers,

We thank the Reviewers for expressing a high level of enthusiasm for our manuscript entitled "MYCN-driven Fatty Acid Uptake is a Novel Metabolic Vulnerability in Neuroblastoma" (NCOMMS-21-29324), and providing insightful comments and suggestions. We have revised the manuscript following editorial requests and addressed the reviewers' concerns point by point below. The changes are highlighted in gray throughout the text.

Reviewer #1:

Comment 1: Line 127: "in two MYCN-induced NB cell line models and in NB primary tumors". One model is Tet-ON Dox-induced, while the other here is conditionally knocked down by Dox-inducible shRNA expression. Perhaps a better way to describe them therefore is "conditionally-expressed MYCN", or other way to reflect that they are not MYCN-induced lines per se.

Response: Thank you for the comment. We revised the text and better described the MYCN systems (line #114).

Comment 2: Line 194: "These findings were validated in a second MYCN system, SK-N-AS MycN-ER." Since this is a new assay (lipid droplets) being introduced to extend the FA phenotype to an additional cell type, why isn't this assay performed in parallel with MYCN3-Tet-ON cells to show that they respond similarly? Fig 2A:

This appearance of lipid droplets after MYCN activation contrasts with the report from Marie Arsenian Henriksson where such droplets were observed after Myc inhibition in neuroblastoma (Zirath et al. 2013). Please briefly discuss this discrepancy.

Response: Per reviewer's suggestion, we performed lipid droplets (LDs) staining in MYCN3 Tet-On cells upon MYCN induction (-/+ 1 μ g/mL DOX for 72 h). To exclude a potential contribution of DOX to the observed phenotype, LDs were also assessed in SHEP (MYCN3 sub-cloned) cells in the presence or absence of DOX (-/+ 1 μ g/mL for 72 h). Similarly, LDs were assessed in SK-N-AS MYCN-ERTM cells upon MYCN activation (-/+ 500 nM 4-OHT for 48 h) and SK-N-AS parental cells in the presence or absence of 4-OHT (-/+ 500 nM for 48 h) to determine potential impact of 4-OHT. Turning on MYCN in MYCN3 cells induces LDs ($p=0.01$) to a less extent compared to SK-N-AS MYCN-ER activated cells ($p<0.0001$, **Fig. R1a-b**). In addition, no significant changes in LDs were observed in SHEP cells treated with DOX and in SK-N-AS cells treated with 4-OHT (**Fig. R1a-b**), suggesting that DOX and 4-OHT do not play a role in LDs accumulation. These data are in accordance with our previous LDs observations. Notably, inhibiting FA synthesis (via A939572) and FA transport (via CB5) attenuated MYCN-driven LD accumulation (**Fig. R1c**), suggesting that FA synthesis and transport contribute to LD formation in the context of MYCN activation. To examine specific lipid categories induced by MYCN, we performed lipidomic profiling in SK-N-AS MYCN-ERTM upon MYCN activation (-/+ 500 nM 4-OHT for 48 h, $n = 4$, **new Fig. 1e**). Consistently with LAN5 shMYCN and MYCN3 systems, MYCN activation upregulated most

Figure R1. Lipid droplet formation in NB cells. a-b. Lipid droplets were assessed by Oil Red O staining in MYCN3 and SHEP cells (-/+ 1 μ g/mL DOX for 72 h)(a), as well as SK-N-AS MYCN-ERTM and SK-N-AS cells (-/+ 500 nM 4-OHT for 48 h)(b). Mean \pm SEM ($n=4$); two-sided unpaired t-test. **c.** Oil Red O staining in SK-N-AS MYCN-ERTM (-/+ 500 nM 4-OHT for 48 h) with and without A939 (40 μ M) or CB5 (5 μ M). Mean \pm SEM ($n=4$); two-way ANOVA with Tukey's multiple comparisons test. Scale bar=20 μ m; A.U=arbitrary units; A939=A939572.

glycerolipids, particularly TGs (FDR < 0.25, **new Fig. 1e**). Because glycerolipids are the major components of LDs, this supports our LDs finding. Lipidomics profiling provides more specific characterization of the lipid categories altered by MYCN and these data were included in the **new Fig.1e**. LDs are dynamic organelles involved in energy regulation, signaling, and lipid metabolism. MYC(N) dynamically alters lipid homeostasis by promoting FA uptake (findings presented in this manuscript), *de novo* lipogenesis, LDs, and β -oxidation to generate ATP. Here we show that activation of MYCN promotes FA uptake and lipogenesis (Fig. 2a and supplementary Fig. 2a), thus resulting in LDs accumulation. Importantly, blocking FA uptake in the context of *MYCN*-amplification reduces LDs and exerts anti-tumor activity (**Fig. R1c**, Fig. 4e–h and Fig. 5f–g). However, inhibition of MYCN in NB also leads to LDs accumulation due to inhibition of FA β -oxidation and redirection of FAs towards *de novo* lipogenesis (Zirath et al. 2013). The relative contributions of FA uptake, synthesis, storage and oxidation to MYCN reprogrammed lipid homeostasis will need to be dynamically assessed in future investigations.

Comment 3: Fig 2D: While the effects of FA supplementation on the MNA cells vs. SHEP are clear, for the SK-N cells with or without activated MycN-ER it is more subtle (although significant). It is a little confusing to represent the same data in a different format here (Fig 2D right panel vs. the other panels), and perhaps drawing the data in the same way may show how the MycN-ER cells are more similar to the MNA cell lines, whereas without Myc-ER they are more similar to SHEP cells (therefore, plot the right panel of Fig 2D in 2 separate line graphs, not as single bar graph).

Response: Thank you for the suggestion. To address also Reviewer #3 comment #2, we repeated cell viability study in SK-N-AS MYCN-ER[™] cells over time (0-6 days) and plotted the results as two separate line graphs as suggested (**new Supplementary Fig. 2e**, left). In addition, we examined the effects of lipid deprivation on cell viability and apoptosis in an extended panel of MNA and non-MNA lines (**new Fig. 2c**). Our results demonstrate that cells harboring *MYCN*-amplification or high *MYCN* activity are more dependent on exogenous lipids.

Comment 4: Line 286: “Supporting that NB cells highly depends” The grammar requires correction here.

Response: The grammar has been corrected.

Comment 5: Fig 4J: There is no legend for the blue and black bars (in fact, they are colored incorrectly and should be red, presumably).

Response: Thank you for the comment. We revised the graph, which is now correctly colored (**new Fig. 4j**).

Comment 6: Line 425-6: “These effects can be rescued by FA supplementation.” From Fig 2D, it seems that there is only a partial rescue, and in fact in IMR32 and MYCN-ER cells, it is a relatively small effect. This should be reflected in the text here (since as written it implies a complete rescue) plus one could speculate as to why there is not a greater rescue with FA supplementation. This extends to Line 230 too (“Collectively, our data indicate that MNA-induced cell growth critically depends on exogenous FAs.”). Again, if it is critically dependent, why isn’t the rescue from FA supplementation more notable? It is a modest – although significant – effect in Fig 2E (right panel) for MYCN-ER active cells vs. inactive.

Response: Thank you for the comment. We revised the text to emphasize that it is a partial rescue. We repeated cell viability study in SK-N-AS MYCN-ER[™] cells over time (0-6 days). MYCN-ER activated cells were more susceptible to lipid deprivation. Moreover, FA supplementation almost completely rescued the viability of MYCN-ER activated cells ($p=0.0001$; **new Supplementary Fig. 2e**). In addition, we examined the effects of lipid deprivation on cell viability and apoptosis in additional MNA (SK-N-BE(2c)) and non MNA (SK-N-AS parental) cell lines. MNA cells were consistently more dependent on exogenous lipids compared with non MNA cells. Moreover, FA supplementation partially rescued cell viability in all the lines tested ($p<0.001$) and completely rescued the deprivation-induced apoptosis in LAN5 and SK-N-BE(2c) cells (**new Fig. 2c**). Overall, our new data suggest that MYCN-driven cells depend on exogenous FAs.

Reviewer #2:

Comment 1: Please show WB on MYCN, proliferation, and apoptosis markers after treatment with the inhibitors in the cell lines. In addition, authors only show changes in MYCN expression in LAN5 and MYCN3 in S. Figure 1A, but not in any other of their cell lines after induction/repression of MYCN.

Response: As suggested, we assessed the protein expression of MYCN, c-MYC, p53, p21, total and cleaved Caspase-3, as well as total and cleaved PARP in MNA (LAN5 and IMR32) and non-MNA (SHEP) cells upon

CB5 treatment (**new Fig. 4k**). Western blots were performed in triplicate and quantifications are shown in the **new Supplementary Fig. 5a**. CB5 preferentially induced cleaved PARP and cleaved Caspase-3 expression in MNA cells (**new Fig. 4k** and **new Supplementary Fig. 5a**). These data are in agreement with the Caspase 3/7 activity analysis shown in Fig 4j. CB5 also inhibited MYCN but not c-MYC protein expression, supporting the selective targeting of MNA cells. p53 and its downstream target p21 play critical roles in cell cycle, proliferation, and apoptosis in the context of *MYCN* amplification. We found that CB5 increased both p53 and p21 expression in MNA cells. Collectively, our data suggest that CB5 inhibits MYCN and activates p53 signaling to suppress cell growth and promote apoptosis. As requested, we added MYCN WB in Tet21/N (Tet-Off) system (**new Fig. 3b**). In the MYCN-ER™ system, 4-OHT induces *MYCN* transcriptional activity. In Fig. 3a we included mRNA expression of *ODC1*, a known MYCN target gene, as positive control for *MYCN* activation. A more extensive validation of the MYCN-ER™ system can be found in our recent manuscript (Moreno-Smith et al., *Nat. Commun.* 2021, doi: 10.1038/s41467-021-24196-4; Fig. 3c).

Comment 2: The authors have analyzed lipid accumulation in some of their experiments. It has previously been shown that inhibition of MYCN results in lipid accumulation see for instance Zirath PNAS, 2013. How does the lipid accumulation occurring after CB5 treatment correspond to the lipid accumulation caused by MYCN inhibition?

Response: In our study, we observed that activation of MYCN promotes FA uptake and lipogenesis, thus resulting in glycerolipids and LDs accumulation (**new Fig. 1c** and response to reviewer #1 comment #2). Importantly, blocking FA uptake via CB5 in the context of *MYCN*-amplification reduces the expression of FA synthesis (*ACC* and *SCD1*) and transport (*FATP2*) proteins (**new Supplementary Fig. 5d**), and thus glycerolipids and LDs accumulation (Fig. 4g-h). It has been previously shown that inhibition of MYCN in NB also leads to LDs accumulation due to inhibition of FA β -oxidation and redirection of FAs towards *de novo* lipogenesis (Zirath et al. 2013; included in revised discussion). The relative contributions of FA uptake, synthesis, storage and oxidation to MYCN reprogrammed lipid homeostasis will need to be dynamically assessed in future investigations.

Comment 3: Figure 4 and legend. Photos of LAN5 and SHEP cells are shown after BOIDPY. It is obvious by eye that the fluorescence is decreasing more in the LAN5 cells but is also clearly decreasing in the SHEP cells. However, in the graph it looks more or less as if there is no decrease in the SHEP cells. In addition, it is indicated that * means significant changes after CB5 treatment and that # indicates significant differences between LAN5 and SHEP cells at same CB5 dose. Please clarify. The use of # is unusual.

Response: Thank you for the comment. To quantify the fluorescence intensity, we took five random fields per treatment group and determined the CTCF as described in the methods section. In the **new Fig 4i**, each white circle indicates the average CTCF from one experiment. The study was repeated three times. We have now included more representative images. We have also separated LAN5 from SHEP graphs for individual statistical analysis, and individual p-values are now presented.

Comment 4: Lines 407-410 “Our metabolomics data show reduced glutamine and glutamate in MYCN-overexpressing cells and MNA tumors, suggesting enhanced consumption of glutamine and glutamate. Future research should include in vivo imaging and flux studies to better understand MYCN reprogramming of glutamine metabolism.” Other studies have demonstrated that MYCN *de novo* glutamine synthesis. Please mention and give reference.

Response: Based on Reviewer #3 comment #6, we revised the discussion section focusing on lipid metabolism. Thus, the above paragraph has been removed from the discussion. However, this reference is now cited in the introduction section (line #64).

Comment 5: The authors confirm that *SLC27A2* is required for NB survival performing shRNA in LAN5 cells. These results should be validated in a second MNA NB cell line. Moreover, to prove that it is a target of MYCN and not c-MYC, authors should also prove that sh*SLC27A2* does not affect viability in non-MYCN NB cells.

Response: Thank you for the suggestion. We genetically depleted *SLC27A2* in a second MNA cell line (IMR32) and in a non MNA cell line (SK-N-AS). We found that silencing *SLC27A2* reduced FA uptake and cell growth in IMR32 but not SK-N-AS cells (**new Supplementary Fig. 4c**), confirming that these phenotypes are selective to MNA cells.

Comment 6: Author demonstrated lipid droplet accumulation as a result of increased fatty acid uptake and synthesis was demonstrated in SK-N-AS MYCN-ER cells. Authors should validate that this is a result due to

MYCN, by repeating this experiment in another cell line, for instance the MYCN3, in which they performed the tracing experiments. This will prove that lipid accumulation is not due to treatment with 4-OHT which could induce stress.

Response: We performed the experiments as suggested, including all the controls. The results are presented and discussed in the response to Reviewer #1 comment #2.

Comment 7: In the Figure legend for Figure 4 E-G, it is mentioned ctrl versus CB5 while the Figure 4 E-G shows ctrl versus sh*SLC27A2*. Please revise.

Response: Thanks for noticing this. We revised the figure legend accordingly.

Comment 8: Line 244. Specify the coordinates of the region analyzed for *SLC27A2* in the ChIP-qPCR experiments. Are there any E-boxes in that region? It would also be useful to check available ChIP-seq data in the ENCODE database and see if c-MYC also binds to the promoter of *SLC27A2*.

Response: Thank you for the suggestion. In the revised manuscript (line #228) we have now included the region of the *SLC27A2* promoter (chr15: 50,182,222-50,182,302) where MYCN binding was assessed by ChIP-qPCR. This region contains a non-canonical E-box (CACCTG) as shown in the **new Supplementary Fig. 3c**. As suggested, we then looked at potential c-MYC binding to the *SLC27A2* promoter (chr15: 50,182,006-50,182,467) in A549 (lung cancer), K562 (leukemia), and MCF-7 (breast cancer) cells, as well as MCF-10A (immortalized breast epithelia) cells (**Fig. R2**). In addition, to definitively assess binding in NB cells, we performed c-MYC ChIP-qPCR analysis in SH-SY5Y NB cells with high c-MYC expression, including *ODC1* (a known MYC(N) target) as positive control. c-MYC binds to the same promoter region (chr15: 50,182,222-50,182,302) but with lower affinity compared with MYCN (4 fold enrichment). These data are included in the **new Supplementary Fig. 3b**.

Comment 9: The authors show a non-canonical E-box in the promoter of *SLC27A2* and show MYCN binding to the promoter. To show that it is a target gene authors need to mutate the E-Box and analyze binding by ChIP. Furthermore, both wt and mutant E-box-reporters, for instance luciferase, and assays should be performed to validate transcriptional activation of *SLC27A2* by MYCN.

Response: To address this comment, we performed site specific mutagenesis and modified the E-box of the luciferase construct carrying the *SLC27A2* promoter. We then assayed luciferase activity as a function of MYCN expression levels in TET/21N Tet-Off system. As shown in the **new Supplementary Fig. 3d**, the mutated promoter is no longer responsive to MYCN, indicating that the identified E-box is a genuine target of MYCN binding activity. The reviewer also suggested to mutate the E-box and then perform ChIP in NB cells. Based on our experience (Gamble et al. 2021. *Cancers*; doi: 10.3390/cancers13081807), this approach is technically challenging and has low chances of success. NB cell lines are defective for homology directed repair, which is a pre-requisite for successful site specific genome editing. We believe that the luciferase assays along with the many reproducible ChIP results we presented should suffice to support the view that MYCN directly associates at the *SLC27A2* promoter through the specific E-box site.

Comment 10: The major concern is the lack of biological replicates in many of the *in vitro* experiments shown in this manuscript. In the materials and methods section, the authors mention that at least six mice are necessary for *in vivo* experiments to reach an 80% statistical power at p<0.05. But for several of the *in vitro* experiments, the authors only present two replicates. Statistical analysis cannot be performed with n<3. Authors should discuss how the statistical analysis was performed and represent experiments with at least three times for significance. It is important to present at least three replicates per experiment with the corresponding statistical

analysis. This concerns the following Figures: Fig 2B (n = 2-3), 2C (n = 2), 2D (n = 2-4), 2E (n = 2), 3C (n = 2), 4D (n = 2), and 6B (n = 2). Supplementary Figures 2B (n = 2-4), 2C (n = 2), 2D (n = 2), 3B (n = 2), 4B (n = 2-3), 6B (n = 2).

Response: We performed original statistics in GraphPad Prism 7.01, which has a FAQ page for n = 2 (<https://www.graphpad.com/support/faqid/591>). We agree with the reviewer that at least three biological replicates should be performed for significance. We performed new studies to obtain at least three biological replicates for each figure panel mentioned above (**new Figs. 2a, 2b, 2c, 3c, 4d, and 6b; new Supplementary Figs. 2b, 2c, 2d, 2e, 3b, 4b, and 6c**). The updated raw data can be found in the source data files.

Comment 11(a): Please show the actual size of the tumors from the different mouse models so it is possible to compare them.

Response: Final tumor weights (g) are presented for all the *in vivo* studies. Actual tumor pictures were taken for one MNA model (LAN5-luc, Fig. 5b) and one non MNA model (SK-N-AS, Fig 5c). These are now presented in the **new Supplementary Fig. 5b–c**, supporting our findings that CB5 inhibits the growth of LAN5 xenografts but does not reduce the tumor size of SK-N-AS xenografts. Tumor pictures were not taken for allograft and syngeneic models because of tissue biochemical and metabolic analyses. However, MRI tumor quantifications are shown for the allograft model in Fig. 5e, supporting final tumor weights (g).

Comment 11(b): Figure 5H. Authors have used mice with normal immune system and write that it is important that tumor growth can be reduced by inhibiting *SLC27A2*. Is it not the growth of the TH-MYCN spheres which is the major achievement in this experiment, ie. that tumors can form even in the setting of a normal immune system rather than that the tumor burden can be reduced? Comparing the graphs in Figure 5F with Figure 5H, respectively, show that the tumors in fact get larger in the syngenic model compared to in the nude mice during the same time frame. Do the authors have any idea why the tumors from the TH-MYCN^{+/+} cells become larger in the 129x1/svj compared to the NRC nude mice? Please comment on this in the discussion.

Response: The goal of our *in vivo* studies was not to compare the growth of MYCN-induced tumors in an immune compromised (allograft model) vs. an immune competent (syngeneic model) background, but to assess the anti-tumor activity of CB5 in these models. Different TH-MYCN^{+/+} tumors (with potential differences in clinical behavior) were used to generate allograft and syngeneic models. Because both source tumor and host are different, a direct comparison cannot be made. We specified that different tumors were used to generate these models in the revised methods section (line #562).

Comment 11(c): The authors have analyzed some of the tumors generated for lipid accumulation but not stained for any other markers (Figure 4G and Figure 5G). Please show staining of the tumors at endpoint for at least MYCN, proliferation, apoptosis and maybe also for hypoxia, and angiogenesis markers to understand more about the mechanism for reduced tumor growth. In addition, how did the authors choose the mice to analyze for lipid accumulation? Please explain why you chose to stain for lipids only in the LAN5 shCTRL and sh*SLC27A2* and in the TH-MYCN ^{+/+} derived allograft model in NCR nude mice.

Response: As suggested by the reviewer, we performed immunohistochemistry (IHC) analysis for MYCN, Ki67, and cleaved Caspase-3 in LAN5 shCTRL and sh*SLC27A2* tumors, as well as in LAN5 tumors after treatment with CB5 as single agent and in combination with VP16 to determine how genetic and pharmacological interference of *SLC27A2* inhibits tumor growth. We found significant reduction of Ki67 and increase of cleaved Caspase-3 levels in sh*SLC27A2* and CB5-treated tumors compared with CTRL tumors ($p < 0.01$, **new Supplementary Fig. 4d-e** and **new Supplementary Fig. 6d-e**), suggesting that blocking FA transport suppresses tumor proliferation and induces tumor apoptosis. The combination therapy CB5+VP16 exerted greater anti-proliferative and pro-apoptotic effects than single therapy ($p \leq 0.05$, **new Supplementary Fig. 6d-e**). Moreover, CB5 reduced MYCN expression in MNA NB tumors ($p < 0.01$, **new Supplementary Fig. 6f**), supporting our *in vitro* data (**new Fig. 4k**). We decided to perform Oil Red O staining in one genetic mouse model (sh*SLC27A2*) and one pharmacological model (CB5 in allografts). We chose the allograft model because it is a MYCN-induced model, which is largely characterized in our manuscript (see MRI quantifications), and nicely responds to CB5, except for two non-responder tumors (Figs. 5d-g). Interestingly, these two tumors did not exhibit lipid inhibition and showed signs of MYCN activation and FA synthesis/transport activity (**new Supplementary Fig. 5d** and Reviewer #3 comment #5).

Comment 11(d): Why did authors wait 2 weeks in the first and then 5,5 weeks in the experiments with the MNA patient tumors?

Response: In the first PDX study (Fig. 5i), mice were treated with CB5 2 weeks after implantation when tumors were not initiated to determine the effect of CB5 on tumor prevention. CB5 did not prevent tumor initiation (Fig. 5i, left). However, chronic CB5 treatment (42 days) significantly prolonged animal survival ($p=0.004$, Fig. 5i, right). In the second PDX study (Fig. 6d), mice were treated with CB5 5.5 weeks after implantation when tumors were initiated (confirmed by MRI) to determine the anti-tumor activity of CB5 alone or in combination with VP16 on established tumors. Animals treated with the CB5+VP16 combination therapy survived significantly longer than those receiving single-drug therapies ($p<0.05$, Fig. 6d).

Comment 11(e): The representation of tumor volume is unclear in Figure 5E. The authors show a significant difference between control and CB5-treated mice, however the graph shows tumor volumes similar for both groups. Please, provide the average of each group.

Response: The MRI images present a single coronal slice for the three-dimensional tumor volume, which is calculated through segmentation of multiple slices (see methods section). Per reviewer's suggestion, the volume average \pm SD of each treatment group is now included in the **new Fig. 5e**.

Comment 11(f): SFigure 5C and SFigure 5D. Why do the mice increase weight during the experiments with treatment with CB5 in TH-MYCN+/+ derived syngenic and in patient derived MNA xenograft models, respectively? Furthermore, no statistics is given in these experiments. The same is true for the experiments shown in SFigure 5A and SFigure 5B.

Response: Thank you for the comment. For all the *in vivo* studies, we re-plotted mouse weight averages as single lines to better appreciate potential changes over time and performed statistical comparison. For all the studies, no significant differences in mouse weights between CTRL and CB5-treated groups were detected at each measured time point (**new Supplementary Fig. 5b-f**), suggesting that CB5 has low toxicity profile in mice.

Comment 11(g): Authors show that CB5 has minimal effect in non-MNA cells *in vitro* but they have not proved this *in vivo*. Treatment of mice implanted with non-MNA NB cells should be performed to further validate the *in vitro* results. Moreover, NCr nude mice were treated with 25 mg/kg b.i.d, while in the immunocompetent 129x1/svj mice the concentration of CB5 was 30 mg/kg b.i.d. Please, explain why the concentration is higher in the latter.

Response: Thank you for the suggestion. To confirm the selectivity of our approach, we tested the effect of CB5 in xenografts derived from non-MNA SK-N-AS cells. CB5 did not reduce tumor volumes and weights of SK-N-AS xenografts (**new Fig. 5c and new Supplementary Fig. 5c**), suggesting that CB5 preferentially inhibits MNA tumors. Our syngeneic model is a much more aggressive model (as discussed in comment #11b), thus we used a slightly higher CB5 concentration.

Comment 11(h): The *in vivo* experiments show that the mice do not lose weight as a sign of toxicity derived from inhibition of *SLC27A2*. In the discussion it is mentioned that normal tissue toxicity is missing while inhibiting FA synthesis. Does inhibition of *SLC27A2* affecting organs as the liver or kidneys?

Response: Thank you for the question. To assess organ toxicity, mouse normal livers and kidneys upon treatment with CB5 as single agent or in combination with VP16 were histologically evaluated by our TCH Pathology Core (blinded pathology review). No evidence of normal organ toxicity was noted. The representative H&E staining images are now included in the **new Supplementary Fig. 6g**.

Comment 11(i): There are important differences between the mice treated with CB5 in Figure 5E but still the differences with the control group are significant at a p-value of 0.006. Please revise and provide a graph with the average and SD of all the mice. The present Figure 5E can be shown in SI.

Response: We reviewed data and statistics (two-way ANOVA with Sidak's multiple comparisons test). At day 14, the p-value between CTRL and CB5 group is 0.006. As suggested, the volume averages \pm SD are now included in the MRI images (**new Fig. 5e**, comment #11e) and the line graph is now moved to the **new supplementary Fig. 5d**. Standard y axis instead of two segmented axes was used to avoid confusion.

Comment 12: To avoid any confusions, authors should mention that *SLC27A* encodes FATP2 earlier in the text, also in the abstract. Along the same line, authors use abbreviations in the results part that are not spelled out. For easier reading of the manuscript is it preferable to spell out the abbreviations in the text rather than in the Figure legends. Some examples: Line 128 MNA and non-MNA, Line 150 KD, OE, Line 185 D2O, 4-OHT. In the animal experiments: b.i.d. Similarly, for the names of the metabolic enzymes: FASN, ACACA, and SCD1.

Response: Thank you for the suggestion. We revised the text as suggested.

Comment 13: The following sentence should be revised, and relevant references cited, as there are many studies evaluating the metabolic changes induced by MYCN in different NB models and, also, in patient samples: “However, no studies have comprehensively evaluated the metabolic changes induced by MYCN across multiple NB models, including patient samples.” Some recent references:

- Wang T, Liu L, Chen X et al. MYCN drives glutaminolysis in neuroblastoma and confers sensitivity to an ROS augmenting agent. *Cell Death Dis* 9, 220 (2018).

- Oliynyk G, Ruiz Perez M, Sainero-Alcolado L et al. MYCN-enhanced oxidative and glycolytic metabolism reveals vulnerabilities for targeting neuroblastoma. *iScience* 21, 188–204, (2019).

- Ding Y, Yang J, Ma Y, et al. MYCN and PRC1 cooperatively repress docosahexaenoic acid synthesis in neuroblastoma via ELOVL2. *Exp Clin Cancer Res.*38(1):498 (2019).

- Ruiz-Pérez MV, Sainero-Alcolado L, Oliynyk G, et al. Inhibition of fatty acid synthesis induces differentiation and reduces tumor burden in childhood neuroblastoma. *iScience* 24(2):102128 (2021).

Response: In our original statement we wanted to stress that there are no studies providing unbiased metabolic profiling in NB primary tumors with and without *MYCN* amplification. We added the missing references and substantially revised the statement mentioned above for clarity (line #112).

Comment 14: The authors do not specify the time point in which the IC50 values in Figure 2C were calculated.

Response: Time point (72 h) is now included in the legend of **new Fig. 2b**.

Comment 15: In line 246, Figure 3B is mentioned while it should be Figure 3C.

Response: Thank you for pointing it out. This was corrected.

Comment 16: In Lines 352 and 353, Figure 5J is mentioned but there is no Figure 5J in Figure 5. Please revise.

Response: The text is now revised.

Comment 17: The authors write “Our study provides the first direct evidence that MYCN-driven tumors rely on FA uptake and lipid synthesis for survival, making targeting of FA uptake a promising therapeutic approach for high-risk MNA patients.” Again, there are several articles also demonstrating this statement, so the words “first evidence” should be avoided. Authors use similar expressions elsewhere in the manuscript, please do not use “for the first time”, “first report” and similar expressions.

Response: Thank you for the comment. We revised this and other similar statements.

Comment 18: Lines 178-180. How do the authors explain the mechanism of MYCN promoting FA desaturation? Does MYCN regulate any enzyme involved in that process?

Response: In our recent publication (Moreno-Smith et al., *Nat. Commun.* 2021, doi: 10.1038/s41467-021-24196-4) we have shown that MYCN directly upregulates the FA desaturation gene Stearoyl-CoA Desaturase *SCD1*.

Comment 19: Figure 6A-B and Supplementary Figure 6A-B. Please show the cell viability graphs for each cell line and treatment. SHEP cells do not show Caspase 3/7 activity, but we cannot know whether these compounds still affect cell viability. There is the possibility that the selected compounds induce cell death in SHEP by a mechanism other than apoptosis.

Response: We revised the presentation of the synergy plots to improve clarity. We have also now presented cell viability graphs for all cell lines tested using drug concentrations matching synergy studies (**new Supplementary Fig. 6a** and **new Supplementary Fig. 6b**) and Caspase 3/7 activity studies (**new Fig. 6b** and **new Supplementary Fig. 6c**). Compared with LAN5 and IMR32 cells, SHEP cell viability was not affected under the same treatment conditions.

Comment 20: Figure 4J. The different concentrations are presented as white (0), grey (5 μ M) and red (10 μ M) CB5, but there are bars in the Figure with blue and black colors. Nothing is neither mentioned about the blue and black bars in the legend. Please advise and revise.

Response: We revised the graph as suggested (**new Fig. 4j**).

Comment 21: The authors are using several different inducible neuroblastoma cells for different assays. Please clarify the reasoning behind using these specific cells for the specific assays and if they have tested also some of the other cell lines in the experiments presented than the ones shown, with similar or with different results.

Response: Thank you for the comment. In the revised manuscript, we performed lipid profiling in SK-N-AS MYCN-ER[™] cells and found robust accumulation of glycerolipids (**new Fig. 1e**). This system was also used for

FA tracing and mRNA expression to determine MYCN-driven FA uptake and regulation of FA transporters. Some of the findings were then confirmed in the MYCN3 Tet-On system. The Tet21/N Tet-Off system was used in our lab for CHIP and luciferase assays (Moreno-Smith et al., *Nat. Commun.* 2021, doi: 10.1038/s41467-021-24196-4). Thus, we used this system to determine MYCN binding to *SLC27A2* promoter and its impact on *SLC27A2* transcriptional activity.

Comment 22: Please use italics when referring to the gene and for in vitro, in vivo, i.p, b.i.d., etc.

Response: This has been revised.

Comment 23: Line 381 “All treatments did not cause significant body weight loss or clinical signs of toxicity.” Revise to -> No treatment caused any significant body weight loss or clinical signs of toxicity.

Response: We have revised this sentence as suggested (line #385).

Comment 24: Figure Figure 3C, SFigure 3B and SFigure 3C. Please specify the regions and the primers used for the PCRs.

Response: This information is now included in the **new Supplementary Data 4**.

Reviewer #3:

Comment 1: Several of the bar graphs shown in the Figures do not show error bars in the controls (i.e. Fig. 2E, 3B etc). Variability of the controls should also be displayed.

Response: Thank you for the suggestion. We updated all the bar graphs showing error bars in the control samples, except for CHIP-qPCR and cell number figure panels. In MYCN CHIP-qPCR analyses (Fig. 3c and **new Supplementary Fig. 3b**) data were normalized by $2^{-\Delta\Delta CT}$ method, with fold-change values calculated relative to corresponding input replicates, thus no error bar is displayed for input control. In the cell number analysis in shCTRL and sh*SLC27A2* conditions (Fig. 4c and **new Supplementary Fig. 4c**) the same number of cells was plated at day 0, thus no error bar is displayed for day 0.

Comment 2: The different sensitivity of SK-N-AS MYCN-ER cells to medium lipid depletion is quite small. The cells without MYC activation are already quite sensitive to lipid depletion (similar to LAN5 cells). Is this explained by a leakiness in the induction system? How would parental SK-N-AS cells respond? Given that these cells are non-MYC amplified, a high sensitivity towards lipid depletion would be contradictory to the authors' conclusions.

Response: Thank you for the comment. To address also Reviewer #1 comment #3, we repeated cell viability study in SK-N-AS MYCN-ER™ cells over time (0-6 days) in complete and delipidized media conditions, and plotted the results as two separate line graphs as suggested (**new Supplementary Fig. 2e**, left). MYCN-ER activated cells are more sensitive to lipid depletion ($p=0.0001$). In addition, we examined the effects of lipid depletion on cell viability and apoptosis in an extended panel of MNA (LAN5, IMR32, and SK-N-BE (2c)) and non-MNA lines (SHEP and importantly SK-N-AS parental cells) (**new Fig. 2c**). Our results demonstrate that cells harboring *MYCN*-amplification are more dependent on exogenous lipids.

Comment 3: The specificity of the inhibitors used in this study is purely based on literature evidence. As a large proportion of the in vivo findings rely on the CB5 compound, it would be good to establish at least some level of specificity of this drug in the systems used here. For example, the authors could monitor restoration of fatty acid uptake after overexpression of another fatty acid transporter that should not be affected by the compound.

Response: We appreciate the suggestion. Because the FATP2 inhibitor CB16.2 also suppresses FATP1 activity (Zhang et al., *Cancer Discovery*, 2018), we verified the specificity of CB5 for FATP2 as suggested. Ectopic overexpression of both FATP1 and FATP2 increased FA uptake in MNA cells. However, CB5 only reduced FA uptake in FATP2-overexpressing cells (**new Supplementary Fig. 4g**), suggesting that CB5 specifically targets FATP2 in NB.

Comment 4: The lipidome analysis of tumour tissue is somewhat confusing. The majority of lipids displayed in the heatmap in Fig. 4H, including phosphatidylcholine species, actually show a strong upregulation in *SLC27A2*-silenced tumours. Only DG and TG species seem to be selectively downregulated. Does this mean that fatty acids taken up by the tumour cells are funnelled specifically into the TG synthesis pathway? How does this compare to the lipid species identified to be altered in the initial metabolomics analysis performed on cell lines and tumours (Fig. 1)? This experiment needs deeper exploration and discussion.

Response: We appreciate the reviewer's observation, and looked into the metabolomics analysis of lipid species. Indeed, MYCN appears to reduce phospholipid synthesis and upregulate glycerolipid synthesis and DG levels (Fig. 1c, Supplementary Fig. 1b). This likely explains why silencing *SLC27A2* in the context of *MYCN*-amplification effectively inhibits MYCN-induced glycerolipid accumulation (mostly DG and TG species; Figs. 5g-h). When looking at the phospholipid and glycerolipid biosynthesis pathways, DGs are derived from phosphatidic acid (PA) by phosphatidic acid phosphatases and converted to TGs by diacylglycerol-acyltransferase1 or 2 (DGAT1 and DGAT2). On the other end, PA can be converted to CDP-DGs by CDS1 and CDS2, leading to other types of phospholipids (PC, PE, PG, PI, and PS). We speculate that MYCN may directly regulate glycerolipid and TG synthesis by upregulating DGAT2 to store excess FAs in TGs and LDs. *DGAT2* is highly expressed in MNA patients across two large datasets (GSE45547 and GSE85047), and high expression predicts NB poor clinical outcomes ($p < 0.01$, **new Supplementary Fig. 7a**). In addition, MYCN upregulates *DGAT2* expression in multiple *in vitro* NB models ($p < 0.05$), and blocking FA transport via CB5 suppresses its expression ($p < 0.05$, **new Supplementary Fig. 7b**). Because glycerolipid and phospholipid biosynthesis share the same precursor PA, a shift towards glycerolipid synthesis may inversely affect phospholipid synthesis. Future investigations remain necessary to elucidate how MYCN dynamically maintains lipid homeostasis. Future studies will aim at tracing FA incorporation into lipids and performing a comprehensive evaluation of the enzymatic activities involved in glycerolipid and phospholipid metabolism. This comment was addressed in the revised discussion.

Comment 5: In the orthotopic allograft model (TH-MYCN+/+ in NCr nude mice, Fig. 5D-G), there are two clear outliers in the CB5 treatment group. Where these tumours also analyzed by Oil-Red-O staining? This would be important to judge whether failure to inhibit tumour growth could be due to poor availability of the drug in the target tissue or due to potential resistance mechanisms (i.e. activation of fatty acid synthesis). Would it be possible to confirm drug availability in the tumour tissue?

Response: Thank you for the comment. Indeed, two tumors escaped CB5 treatment (Fig. 5f). Supporting our findings, these tumors did not exhibit lipid inhibition by Oil Red O staining (their neutral lipid level is comparable to CTRL tumors, **new Supplementary Fig. 5d, bottom**). Unlike CB5-responsive tumors, these tumors also showed increased MYCN activation and FA synthesis/transport activity (*SCD1* and *FATP2*, **new Supplementary Fig. 5d, middle**). CB5 likely fails to inhibit MYCN and MYCN-mediated FA transport and synthesis in these two tumors, resulting in LDs accumulation.

Comment 6: The manuscript is quite difficult to read. This is most likely due to that large amount of data but some editing for clarity would help making the study more accessible. In particular, the first paragraph of the discussion section should be reconsidered. The authors mention several isolated findings and speculate about potential mechanisms that were not investigated in the manuscript. This distracts from the discussion of the main conclusions from the study. In particular, the discussion of potential roles of polyamine and glutamine metabolism in MYCN-amplified NB does not add further insight. Moreover, the discussion refers to the downregulation of glycerophosphoglycerols by MYCN but this is not mentioned in the results section and cannot easily be delineated from the figures. The section on metabolism and immune function is also quite unclear as the authors report reduced tumour growth in both immunocompetent and deficient *in vivo* systems.

Response: We appreciate the suggestion. We substantially revised our manuscript language to improve clarity. The first paragraph of the discussion was revised and is now focused on lipid/DG pathway as suggested, and the section of metabolism and immune function was clarified. We believe our revised manuscript is much improved in terms of clarity and focus.

Comment 7: Line 101: Something seems to be missing from this sentence. Should this be in a new paragraph?

Response: Thank you, this sentence is now revised (line #89).

Comment 8: Line 128: The description of the experimental systems used in this study is a bit confusing. It should be made clear that MYCN overexpression (MYCN3 model) was performed in non-MYCN amplified SHEP cells.

Response: We appreciate the comment. The MYCN models, including MYCN3, are now described in more detail (line #120 of the results section; see also Reviewer #1 comment #1).

Comment 9: Figure 1C: It would be useful to assign the indicators of significance to the respective comparison (i.e. tumours, overexpression or KD). Asterisks could be organised in a grid. It would be particularly interesting to see pathways that are only significantly regulated in tumours. Remarkably, only one pathway (mapping to DG metabolism) is significantly altered in all three experimental systems. This should be discussed.

Response: Thank you for the comment. We decided to arrange the asterisks inside the circles that indicate the number and abundance score of metabolites within each subpathway, hoping this will clarify which comparisons are significant. Unbiased metabolomics suggests that the DG pathway is the only pathway consistently altered in all three experimental systems (Fig. 1c–d). Lipidomics profiling then confirmed that DGs and TGs are consistently upregulated by MYCN in all three systems (Fig. 1e), supporting the importance of glycerolipid accumulation in MNA NB. As discussed in the revised discussion section, we found that targeting FA transport effectively inhibits MYCN-induced glycerolipid accumulation and tumor growth, suggesting that FA transport is critical for these functions (Fig. 4 e-h). However, glycerolipid synthesis may also be directly promoted by MYCN via upregulation of DGAT2 (**new Supplementary Fig. 7**) as discussed in comment #4 and revised discussion.

Comment 10: Figure 1E: Why were these TG species (TG 42:0 and TG 48:2) chosen for display as box plots?

Response: We originally thought to show representative TG species upregulated by MYCN across different MYCN systems. To better elucidate the link between FAs and glycerolipids, we decided to analyze the FA chain composition of MYCN-induced glycerolipids across the three systems. The FA chains 14:0, 16:0, 16:1, and 18:1 were the most represented (>30%, **new Supplementary Fig. 1c**). Consistently, MYCN increased these FA levels in NB cells and tumors (Supplementary Fig. 1d–e), suggesting that MYCN upregulates the abundance of FAs required for glycerolipid accumulation.

Comment 11: Line 154: GSEA on the same data cannot be used to “validate” findings. This should be rephrased.

Response: Thank you for the comment. This has been revised (new line #142).

Comment 12: Line 205: The statement that both A939572 nor CB5 have no cytotoxic effect but that CB5 is less toxic does not make much sense. This should be rephrased.

Response: This part has been revised (line #190).

Comment 13: Line 246: Reference to figure should be 3C.

Response: Thank you. This has been corrected.

Comment 14: Line 317: The MNA LAN5 orthotopic implantation model has already been introduced on the previous page.

Response: This has been revised.

Comment 15: Line 453: The meaning of this sentence is unclear. Do the authors mean immune cell activity targeted against MYCN positive NB tumours or MYCN function in immune cells.

Response: Thank you for the comment. This has been revised (line #443).

Comment 16: Line 455: Insert new paragraph after “therapeutic targets”?

Response: Thank you. This has been revised.

Comment 17: Typo in temozolomide.

Response: Thank you. This has been corrected.

REVIEWERS' COMMENTS

Reviewer #1 (Remarks to the Author):

The authors have addressed all the comments and added a significant amount of new data to the paper. It is a nice piece of work now ready for publishing.

Reviewer #2 (Remarks to the Author):

The authors have made a tremendous work in addressing the concerns from the three referees. This effort has made the ms very good and it is now well suited for publication in Nature Communications.

Reviewer #3 (Remarks to the Author):

The authors have addressed all issues and have substantially improved the manuscript. It can now be considered for publication.